nanotechnology/environmental science

greenhouse emissions, nanomaterials, metal oxide semiconductors, gas sensors

**Authors for correspondence:**
Yogendra K. Gautam
e-mail: ykg.iitr@gmail.com
Kavita Sharma
e-mail: sharmak29@gmail.com
Beer Pal Singh
e-mail: drbeerpal@gmail.com

This article has been edited by the Royal Society of Chemistry, including the commissioning, peer review process and editorial aspects up to the point of acceptance.

# Nanostructured metal oxide semiconductor-based sensors for greenhouse gas detection: progress and challenges

Yogendra K. Gautam, Kavita Sharma, Shrestha Tyagi, Anit K. Ambedkar, Manika Chaudhary and Beer Pal Singh

Smart Materials and Sensor Laboratory, Department of Physics, CCS University, Meerut, Uttar Pradesh 250004, India

(iD) YKG, 0000-0003-4662-4583; KS, 0000-0002-9054-0208; BPS, 0000-0002-1646-8404

Climate change and global warming have been two massive concerns for the scientific community during the last few decades. Anthropogenic emissions of greenhouse gases (GHGs) have greatly amplified the level of greenhouse gases in the Earth's atmosphere which results in the gradual heating of the atmosphere. The precise measurement and reliable quantification of GHGs emission in the environment are of the utmost priority for the study of climate change. The detection of GHGs such as carbon dioxide, methane, nitrous oxide and ozone is the first and foremost step in finding the solution to manage and reduce the concentration of these gases in the Earth's atmosphere. The nanostructured metal oxide semiconductor (NMOS) based technologies for sensing GHGs emission have been found most reliable and accurate. Owing to their fascinating structural and morphological properties metal oxide semiconductors become an important class of materials for GHGs emission sensing technology. In this review article, the current concentration of GHGs in the Earth's environment, dominant sources of anthropogenic emissions of these gases and consequently their possible impacts on human life have been described briefly. Further, the different available technologies for GHG sensors along with their principle of operation have been largely discussed. The advantages and disadvantages of each sensor technology have also been highlighted. In particular, this article presents a comprehensive study on the development of various NMOS-based GHGs sensors and their performance analysis in order to establish a strong detection technology for the anthropogenic GHGs. In the last, the

scope for improved sensitivity, selectivity and response time for these sensors, their future trends and outlook for researchers are suggested in the conclusion of this article.

## 1. Introduction

The whole world is craving for an environment on Earth plenteous of clean and fresh air. It ought to comprise reduced levels of greenhouse gases (GHGs); carbon dioxide ($CO_2$), methane ($CH_4$), nitrous oxide ($N_2O$) and ozone ($O_3$) and fluorocarbons. Human activities have increased the level of GHGs in the Earth's environment in dramatic ways over the past two centuries. Since the dawn of the industrial revolution in the early 1800s also 1900s, the burning of fossil fuels has greatly amplified the level of GHGs in the atmosphere, especially the $CO_2$. Deforestation is considered as the second largest anthropogenic source of $CO_2$ to the atmosphere [1]. The GHGs act like a blanket that absorbs infrared (IR) radiation and prevent it from escaping into outer space. This results in the gradual heating of Earth's atmosphere and surface and the process is named as global warming. The scientific community implicates that global warming might severely damage the Earth's atmosphere and climate.

Currently, most of the developing countries are struggling with the unstoppable excessive rate of change of climate in comparison to developed countries. The climate is changing and it is changing more quickly than is realized by us. According to a recent 'Global Warming of 1.5°C' special report of the Intergovernmental Panel on Climate Change (IPCC), the world has much less time before climate change becomes unmanageable. This report says that 'climate-related risks for natural and human systems are higher for global warming of 1.5°C than at present but lower than at 2°C' [2]. The IPCC fifth assessment report confirms that the total anthropogenic GHG emissions have continued to increase around 1970–2010 with larger absolute increases between 2000 and 2010 and still increasing, despite the growing number of climate change mitigation policies. It is quite evident that more than half of the observed increase in global average surface temperature from 1951 to 2010 was caused by the anthropogenic increase in GHG concentrations [3].

The scientific community has accepted the challenge of changing climate and is rigorously working towards the measurement and the quantification of this change. The prominent factors responsible for the volatility of the environment to a major extent are the GHGs. Diversified sensor technologies have been developed by various research groups worldwide for the detection of these environmentally hazardous gases. The existing sensor technologies are efficiently trying to develop suitable sensors for the precise monitoring of the environment deteriorating GHGs. The different gas detection technologies include mainly metal oxide semiconductor (MOS)-based sensors, catalytic gas sensors, electrochemical gas sensors, optical gas sensors and acoustic gas sensors, etc. The performance of every sensor is based on some characteristic including sensitivity, selectivity, detection limit, operating temperature, response time and recovery time, etc.

Among many gas sensing technologies, nanostructured metal oxide semiconductor (NMOS) based gas sensors have shown excellent performance over other available sensors because of their excellent physical and chemical properties and unique structure [4–10]. These materials have a wide band gap, allowing them to have a full spectrum of electronic properties. The properties of the NMOSs are greatly affected by the material's size, microstructure and moreover the inclusion of some impurities like metals and ions etc., drastically improves their electrical/optical properties [11]. NMOS-based sensors have been shown to be sensitive to a large range of GHGs, mainly to $CO_2$, $CH_4$, $N_2O$ and $O_3$, with excellent responses varying with both target gas concentration and device operating temperature [12]. The past four decades have seen incredible research in the development of MOS-based gas sensors which includes the production of innovative materials and associated fabrication technologies, and of course improved efficiency of sensors [13]. These studies lead to the good quality and efficient portable gas sensing devices, which have shown excellent sensitivities and selectivity; efficient fast response/recovery time; low operating temperature or even the ability to function independent of temperature which would lead to efficient power consumption; stability in the performance of the sensor in environment conditions; and minimum use of the chemically sensitive layer. All these features would allow the sensors to be used a multiple number of times for measurements [14]. However, there is a need for further improvements in their performances such as accuracy, selectivity and reliability in order to meet today's requirement of precise monitoring of the GHGs [15]. There have already been a lot of studies carried out by researchers globally on the topic of

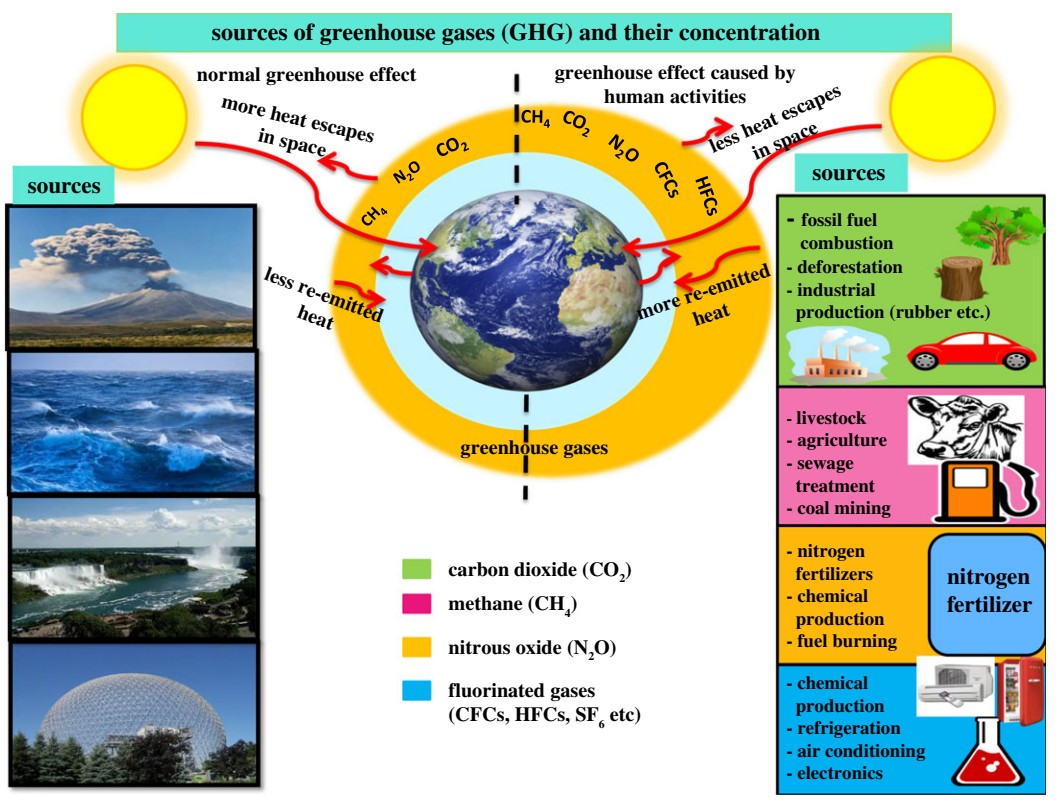

**Figure 1.** A comparative diagram of natural greenhouse gases and the variety of human activities contributing towards the increasing concentration of greenhouse gases in the Earth's environment.

NMOS gas sensing materials in general and few of them excellently describe the gas sensing mechanism in depth. These studies conclude several strategies for the further improvement in the gas sensing properties of the NMOS sensors from the perspective of gas sensing mechanisms [16–18].

In the next part of the introduction, we briefly discuss the current concentration of the major GHGs present in the troposphere of the Earth's atmosphere and reported by several research agencies worldwide and their effects on human life with a view to emphasize the need for accurate detecting/ measuring NMOS-based tools for the GHGs.

## 1.1. Concentration of greenhouse gases in the Earth's environment

GHGs trap the Earth's emitted radiation, which might escape back to space and when these gases are present in the appropriate amount in earth's environment, keep the earth warm enough to sustain life on it. The primary GHGs in the atmosphere are water vapour, $CO_2$, $CH_4$, $N_2O$ and $O_3$. GHGs specifically trap photons of wavelengths in the IR region and therefore act as important temperature regulators of our atmosphere. These gases differ in their ability to absorb energy, that is, they have various radiative efficiencies. They also differ in their atmospheric residence times. Water vapour accounts for by far the largest greenhouse effect (90–85%) because water vapour emits and absorbs IR radiation at many more wavelengths than any of the other GHGs, and there is much more water vapour in the atmosphere than any of the other GHGs. Although water vapour content is highly variable to measure and owing to lack of proper measurements of global water vapour, it is not predictable how much content of water vapour gives rise to the greenhouse effect that in turn affects the temperature of Earth. GHGs have increased greatly since the preindustrial times owing to human various actions as can be seen in figure 1.

The emission of anthropogenic $CO_2$, as an end product of carbon-fuel combustion, is mainly related to the consumption of fossil fuels, construction materials, minerals and industrial materials. The water solubility of $CO_2$ decreases with the increase in temperature of water and owing to this attribute more $CO_2$ given off into the atmosphere. However, owing to its high solubility and reactivity, $CO_2$ allows ready exchange of itself between oceans and atmosphere. $CO_2$ has a very long residence

time in the atmosphere, its emissions cause increases in atmospheric concentrations of $CO_2$ that will last thousands of years [19].

The present global concentration of various GHGs in the environment along the economic sector wise contribution is listed in table 1 and shown in figure 2. As is evident from table 1, $CO_2$ is considered to be the biggest contributor for climate change as it is present in the highest amount in the Earth's environment, i.e. 76% [21]. The power sector may be considered as the major contributor to its high level of concentration and the industrial sector emerges as the second major contributor. The global concentration of $CH_4$ is 16%. This is emitted from the production and transport of coal, natural gas and oil. $CH_4$ emission also takes place from the decomposition of organic waste in agriculture, in municipal solid waste, landfills and the raising of livestock. The contribution of $N_2O$ emission is 6% to the global GHG emissions. This is supposed to be emitted during agriculture and industrial activities, as well as during the combustion of solid waste and fossil fuels. The chloroflorocarbon (CFC) contribution is of 2% to the global GHG emission.

In order to have a clear picture on the causes of emission of major GHG's and their role in the global greenhouse effect starting from $CO_2$, $CH_4$, $N_2O$, $O_3$ to CFCs are briefly discussed in the next subsections.

### 1.1.1. Concentration of carbon dioxide gas

According to the World Meteorological Organization's Greenhouse Gas Bulletin, universally the middle value of the concentration of $CO_2$ is found to arrive at 407.8 parts per million (ppm) in the year 2018, whereas it was 405.5 (ppm) in 2017. This trend uncovers a consecutive decline of the atmosphere owing to the $CO_2$ concentration level. This increment of $CO_2$ concentration level is attributed to fuel burning, backwoods fires, volcanic emissions and unpredictable natural mixes. The clear decision to combat this issue is to lessen the use of petroleum products, incrementing the use of non-polluting powers, start forest preservation endeavours and prevent volcanoes from ejecting [22,23].

### 1.1.2. Concentration of methane gas

$CH_4$ is another significant gas promoting the greenhouse effect. Its expansion is incredibly changing the climatic condition. It assimilates the sun's heat and warms the environment. It is considered to be multiple times more intense than $CO_2$. It is created by the common deterioration of rice paddies, bogs, the guts of creatures, the spoiling of waste and the dissemination of petroleum products like coal, oil or gas [24]. $CH_4$ is broadly used for power generation, hydrogen and ethylene production, and domiciliary warming. It is exceptionally unpredictable in nature and when blended in with the air, may handily cause an eruption in closed zones. Early identification of the existence of $CH_4$ gas is critical to prevent blasts in mechanical and household applications [25].

### 1.1.3. Concentration of nitrous oxide gas

The anthropogenic sources of $N_2O$ emission include; agriculture fields, fuel combustion, wastewater management and industrial processes. On the whole, these are increasing the amount of $N_2O$ in the atmosphere. The natural source of $N_2O$ emission in the Earth's atmosphere is its nitrogen cycle. In the nitrogen cycle, there is the natural circulation of nitrogen among the atmosphere, plants, animals and microorganisms that live in soil and water. These microorganisms break down nitrogen in soils and the oceans. The $N_2O$ molecules may reside in the atmosphere for an average of 114 years before being removed/destroyed through chemical reactions. The aftermath of 1 pound of $N_2O$ molecules on warming the atmosphere is around 300 times that of 1 pound of $CO_2$ [26]. Globally, almost 40% of the total $N_2O$ emissions come from various human activities [27].

### 1.1.4. Concentration of fluorocarbons

The group of fluorocarbons, as is described in the Kyoto Protocol, can be referred as hydrofluorocarbons (HFCs), perfluorocarbons (PFCs) and sulfur hexafluoride (SF6) [28]. The group of HFCs is particularly broad. The concentration levels of these fluorocarbons have increased substantially over recent decades. Their contribution to climate forcing is currently still limited, although steadily increasing from 0.5% in 1990 to 0.8% in 2004 and 1.5% in 2017. Their contribution is expected to continuously increase in the near future because of the long lifetimes (greater than 1000 years in some cases) and the increase in the emissions of new HFCs, such as HFC-134a [29].

**Table 1.** Current concentration of greenhouse gases in the environment along with their sector wise contribution [20].

| s. no | greenhouse gases (GHGs) | total concentration of GHGs in atmosphere (%) | sector wise contribution in GHG concentration | | | | | |
| --- | --- | --- | --- | --- | --- | --- | --- | --- |
| | | | power sector (%) | industrial sector (%) | transportation sector (%) | commercial and residential sector (%) | agriculture sector (%) | other energy (%) |
| 1. | $CO_2$ | 76 | 25 | 21 | 14 | 6 | 24 | 10 |
| 2. | $CH_4$ | 16 | | | | | | |
| 3. | $N_2O$ | 6 | | | | | | |
| 4. | fluorinated gases | 2 | | | | | | |

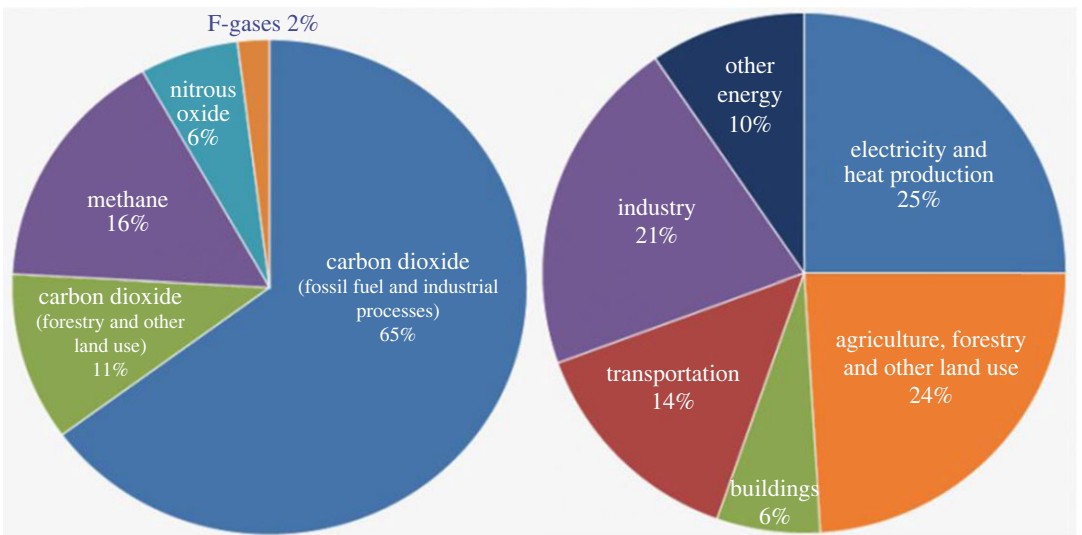

**Figure 2.** The current global concentration of greenhouse gases in Earth's environment along with their global economic sector wise contribution [20].

The status of GHGs in the environment emphasizes the superabundant need for the centralization of these gases in the Earth's environment with the end goal of developing sensors.

## 1.2. Severe impacts of greenhouse effect on human life

Figure 3 depicts the schematic view on some of the severe consequences of increased concentration of GHGs in the Earth's environment created by various human activities such as burning fossil fuels, industrial production (rubber, plastic, etc.), deforestation, sewage treatment, air conditioning, refrigeration, etc. As is evident from figure 3, the first direct well-known effect of the increased concentration of GHGs is the increase in the global average temperature of the Earth. According to the global climate report issued by National Oceanic and Atmospheric Administration, USA. (NOAA), the first three months of the current year 2020 were characterized by warmer-than-average conditions across much of the globe. The January–March 2020 temperatures for Europe and Asia were expected to be the highest in the 111-year record [30]. This has far-ranging environmental and health impacts on living conditions of human life of our planet. Some of the major issues imposed by this are discussed below in short.

### 1.2.1. Ozone layer depletion

It is known that the $O_3$ layer on the Earth protects us all from the sun's harmful radiation, but the increased concentration of GHGs have damaged this shield. Less $O_3$ layer means low protection from ultraviolet (UV) light which may over the time, damage crops and lead to higher skin cancer and cataract rates. The $O_3$ hole is actually a region of exceptionally depleted $O_3$ in the stratosphere prominently over the Antarctic region or the South Pole as is reported in the Scientific Assessment of Ozone Depletion 2014. Primarily it was noted in the beginning of 1970s. On the basis of historical record, it is found that the total column $O_3$ values of less than 220 Dobson Units were not observed prior to 1979. So, 220 Dobson Units of $O_3$ is generally used as the boundary and the values below this can be used for representing the $O_3$ loss. This report also suggests that the $O_3$ depletion is not limited to the area over the South Pole but occurs over the latitudes that include North America, Europe, Asia, and much of Africa, Australia and South America. The ozone depleting-substances (ODS) include CFCs, hydrochlorofluoro carbons (HCFCs), carbon tetrachloride and methyl chloroform. Although ODS are emitted at the Earth's surface, they are eventually carried into the stratosphere in a process that can take as long as 2–5 years [31].

### 1.2.2. Change in rain patterns

Changes in global precipitation are among the most important and least well-understood consequences of climate change or increasing GHG concentrations. Climate change is altering the rain fall patterns

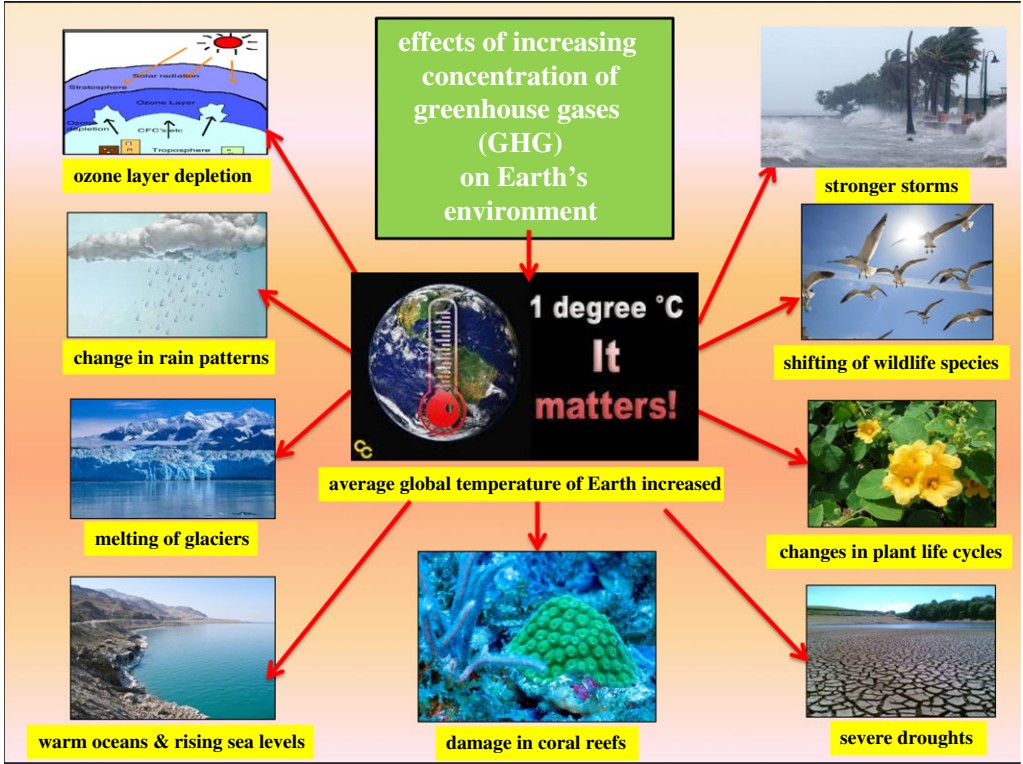

**Figure 3.** A schematic diagram of major severe consequences of increased concentration of GHGs in earth's environment.

globally. According to research [32] climate change affects the global precipitation patterns and shifts them across the land and ocean. The climate model of this research predicts that the addition of heat-trapping gases in the atmosphere will shift precipitation in two main ways. The first shift is in a strengthening of existing precipitation patterns. This is commonly called wet get wetter, dry get drier. The second shift is a change in storm tracks, which should move away from the equator and toward the poles as atmospheric circulation changes. Increases and decreases in the frequency and magnitude of river flood events may vary by region.

### 1.2.3. Melting of glaciers and rising sea levels

Global researchers in recent years have found and reported that the world's glaciers are now in the smallest shape they have been in human history [33]. Latest research quantifies how much the world's lost glaciers have contributed to rising sea levels for example; the sea levels rise by 3 mm each year and the oceans warm further. Scientists estimate that this thermal expansion will force sea levels up even more. It is also claimed that some glacial regions in Europe, Canada, the US and New Zealand could see their glaciers completely disappear by 2100. This fact is also stated in the fifth Assessment Report of the Intergovernmental Panel on Climate Change (IPCC, 2014). According to this report, during the period 1901–2010 the global average sea level rose 19 cm. It is estimated that by 2100 the sea level will be between 15 and 90 cm higher than it is now and will threaten 92 million people [34]. Including this, many extreme temperature conditions are becoming more common. Since the 1970s, unusually hot summer days (highs) have become more common over the last few decades in various parts of globe. Record-setting daily high temperatures have become more common than record lows. The decade from 2000 to 2009 had twice as many record highs as record lows [34].

### 1.2.4. Damage to coral reefs

Coral reefs are considered as one of the most diverse ecosystems on this planet. They offer a surplus of benefits both to natural ecosystems and to the human population. Coral reefs bring in enormous funds to coastal countries through tourism, fishing and discoveries of new bio chemicals and drugs, and additionally they provide natural coastal protection and building materials [35]. However, coral reefs

are experiencing massive die-outs all around the world. At first, many thought the biggest threats to coral reef health were direct anthropogenic effects such as water pollution and sedimentation, but now it is clear that the 50–70% of coral reefs are directly affected by anthropogenic global climate change [36]. Rising global temperatures, increasing oceanic $CO_2$, and other consequences of climate change are all affecting coral reef health in a negative way.

### 1.2.5. Stronger storms

Hurricanes are the most violent storms on the planet, and strong storms are getting stronger because of the warmer oceans. Stronger storms are known to us by different names, for example hurricanes in the Atlantic Ocean, typhoons in the western Pacific Ocean or cyclones in the Indian Ocean. These storms arise owing to a small atmospheric disturbance located in or near a tropical ocean. If water temperatures are warm enough, and atmospheric conditions are supportive with moisture and uniform winds, a tropical system can evolve. In the Atlantic, the system first becomes a tropical depression. As it gets stronger the system matures to a tropical storm and then finally, when winds rise over 74 mph, it turns into a hurricane. Climate scientists claimed that slower steering currents resulting from a warmer climate may have contributed to cyclonic activity [37]. According to this report, the tropical storm activity in the Atlantic Ocean, the Caribbean and the Gulf of Mexico has increased during the past 20 years.

### 1.2.6. Shifting of wildlife species

Change in the Earth's temperature and other impacts of climate change are severely affecting animal species. The world has witnessed many of them go extinct every year owing to changing ecosystems and habitat loss, particularly the tigers, the giant panda bears, green turtles, Asian elephants, polar bears and penguins among others as is stated by Marike Lauwrens in a blog [38]. Climate change affects animal species in a number of ways; the changing climate has made their habitat less comfortable, and sometimes even inhospitable. They have to deal with increases in water, air and solid waste pollution that affects the food they eat and the surroundings they live in. Animals also experience habitat loss because these animals have to alter their breeding and feeding patterns in order to survive.

### 1.2.7. Change in plant's life cycle

The largest known economic impact of climate change is upon the agriculture productivity and livelihood of the large population living in developing countries [22]. Owing to global warming, the expected decline in agricultural productivity is assumed to be by 9–21% in developing countries [39]. In northern and western parts of Asia, rice and maize productions have been reduced to great extent, with an increase in $CO_2$, $N_2O$ and $CH_4$ gas emissions by 5% from 1998 to 2011 [40]. Research studies concluded that a small increase in temperature had larger effect than elevated $CO_2$ on grain quality and the rising trend of global warming is considered to be more striking than precipitation over the twentieth century [41]. The rise in $CO_2$ concentration has both positive and negative effects on agricultural productivity. Through increased photosynthesis, $CO_2$ is predicted to have beneficial physiological effects. C3 crops such as wheat and rice have a higher impact of increased $CO_2$ than C4 plants such as maize and grasses. Above the threshold level of $CO_2$, the hinderance with the respiration system of plants leads to slow metabolic processes resulting in low productivity.

### 1.2.8. Droughts

Global warming is having a profound impact on the processes of soil degradation and is contributing to the desertification of the driest areas on the planet. Desertification destroys all the biological potential of affected regions, turning them into barren and unproductive land. As recognized by the United Nations (UN) on the occasion of the World Day to Combat Desertification in 2018, 30% of land has been degraded and lost its real value [42].

Apart from this, the UN's Food and Agriculture Organization (FAO) states that climate change is raising serious doubts about food availability in its report on the state of world food and agriculture. It warns that a decline in agricultural production would result in food shortages, most severely affecting sub-Saharan Africa and South Asia. Moreover, the World Health Organization (WHO) states

that global warming will cause infectious diseases such as malaria, cholera or dengue to spread to many more areas of the planet. On the other hand, extreme heat will increase and aggravate cardiovascular and respiratory problems [43].

In the view of the above facts and alarming signs, it is the need of the hour to monitor/detect the anthropogenic emissions of GHGs in Earth's atmosphere with utmost priority worldwide. Here, monitoring alludes to the confirmation of the existence of GHGs and their quantification. Therefore, this review article presents a comprehensive analysis of the various studies available on NMOS-based GHG sensors for $CO_2$, $CH_4$, $N_2O$ and $O_3$ along with the factors which affect their sensitivity, selectivity and the stability. Furthermore, the performance of NMOS-based GHGs sensors is critically reviewed and compared with recent studies on other gas sensing techniques such as optical gas sensor, chemiluminescence, photoacoustic spectroscopy, gas chromatography and electrochemical sensors, etc. In the next section of this review paper, different gas sensing technologies have been widely discussed, with a special highlight on MOS-based gas sensors and its sensing mechanism being elaborated in detail.

# 2. Gas sensors and their classification

A gas sensor is a device that detects the presence of the volatile substances in vapour phase, both qualitatively and quantitatively (concentration) in a specific volume [44]. The gas sensors are classified on two different basis: (i) type of sensing material: optical absorption [45], catalytic, thermo conductive [46], solid electrolytic [47] and MOSs [48,49], and (ii) sensing mechanism: gas sensors categorize on detection method is additionally divided into two groups; (a) variation in electrical parameters and (b) sensing mechanism based on other types of alterations like; optical, catalytic, thermometric, photoacoustic, chemiluminescence and gas chromatographic, etc. Materials, such as metal, MOSs, polymers and carbon-based materials are used as gas sensors depending on the change in electrical properties in the presence/absence of target gases [50].

## 2.1. Chemical gas sensors

Chemical sensors were defined by the International Union of Pure and Applied Chemistry (IUPAC) in 1991. A chemical sensor is a device that transforms chemical information, ranging from concentration of a specific sample component to total composition analysis, into an analytically useful signal. It is considered as the official definition of these sensors [51]. There is a change in properties like composition, pH, concentration, etc. In these sensors, effective signal recognition, reception and transduction form the base of quantitative investigation using chemical sensors.

## 2.2. Optical gas sensors

The optical gas sensors are based on the detection of change in physical properties such as intensity, emission spectra, colour, polarization and velocity of light, etc., which are caused by absorbance, reflectance, fluorescence or scattering of light of a particular wavelength by the gaseous species [52]. The use of an optical gas sensor is limited owing to high cost and difficulty in miniaturization [50].

## 2.3. Electrochemical gas sensors

This type of gas sensor consists of electrochemical cells that are made from at least two electrodes, one is a sensing electrode or working electrode and the other is a reference electrode and these electrodes are connected through a thin layer of electrolyte. Electrochemical sensors permit the diffusion of gases through a membrane to the working electrode where oxidation or reduction of the gases takes place [52]. The electrochemical sensors can be divided in three groups: (i) potentiometric sensors, (ii) amperometric sensors, and (iii) conductometric sensors. In potentiometric electrochemical sensors, the resultant potential of the electrode or membrane is measured while in amperometric sensors, the variation in resulting current is measured. On the other hand, for conductometric sensors, frequent series of conductivities of the electrode are measured.

## 2.4. Catalytic gas sensor

Catalytic sensors have been used for the detection of combustible gases for almost a century. The majority of metal oxides and their compounds have catalytic properties. Combustible gas mixtures do not burn until they have reached a certain ignition temperature, but the gas will start to combust even at lower temperatures in the presence of a specific chemical process. This procedure is known as catalytic combustion. The sensor based on this catalytic principle is called a catalytic gas sensor. To test the output of the catalytic gas sensor, the Wheatstone bridge is used. Pellistor and thermoelectric are the two types of catalytic gas sensor [53].

## 2.5. Mass sensitive gas sensor

In the mass sensitive sensor, the target gas mounts on the sensitive adsorbing layer because of which the mass of the sensor surface changes. Surface acoustic wave (SAW) based sensors micro-cantilever and quartz crystal microbalance (QCM) are examples of mass sensitive sensors. The variation in mass shows the change in the properties of sensitive material [54]. The first mass sensitive sensor was based on the estimation of bulk acoustic waves (BAW) in a piezoelectric quartz gem resonator which is sensitive to mass changes. In these types of sensors, acoustic sensors are widely used because their detection mechanism is a mechanical or acoustic wave. When the acoustic wave propagates through the material, a notable variation in the characteristics of the wave (amplitude/velocity) propagation is observed. Changes in velocity can be observed by estimating the frequency and phase characteristics of the sensor and can then be correlated to the corresponding physical quantity being measured. The receptor and transducer are the main components of an acoustic wave sensor. A receptor is an element which is sensitive to an analyte while the transducer is an element which converts the response into an electrical signal [53].

## 2.6. Magnetic gas sensors

The change in the paramagnetic properties of the analyte is the basic principle of magnetic sensors [55]. These sensors are represented by a certain type of oxygen monitor because oxygen has a high magnetic susceptibility as compared to other gases [56]. Therefore, a gas sensor based on the paramagnetic principle and Pauli's exclusion principle for oxygen can be built with only minor cross sensitivities.

## 2.7. Thermometric gas sensors

In thermometric sensors, chemical reactions take place after the interaction of gases with the surface layer of the sensor; these chemical reactions become the cause of variation in temperature. This variation is indicated as the variation in electrical signals such as current, voltage and resistance. It has a temperature probe coated with a chemically selective layer. The device operates by detecting the heat transfer during the catalytic reaction between the coating at the sensor's surface and analyte and the related variation in temperature inside the device is measured [52].

## 2.8. Photoacoustic spectroscopy

Gas phase spectroscopy is now very popular in a wide range of fields such as atmospheric, chemistry, biology, medical sciences and safety. This spectroscopy is based on the photoacoustic effect which was first reported by Alexander Graham Bell [57]. The spectroscopic gas sensors have proved to be invaluable tools. There are various ways of using gas sensors, and different demands are placed on each application. For one particular gas compound, some applications need very high sensitivity, while others benefit more from a sensor that can calculate a wide range of gases or benefit from a miniaturized footprint. There is also a desirable time resolution, as well as selectivity, robustness and little or no need for sample preparation and maintenance; a significant number of these criteria are fulfilled by photoacoustic spectroscopy and its sensors [58].

## 2.9. Chemiluminescence

Chemiluminescence (CL) is one of the luminescent phenomena that can be described as the emission (UV, Vis or IR) emitted by a chemical reaction. In recent years, there has been a growing development in the application of CL to chemical analysis, particularly. As an analytical tool, CL detection is well known for its high sensitivity, quick response and absence of unwanted luminescence in the background. It can occur in the gas, liquid or solid phase, thus facilitating and expanding its scope of analytical applications [59].

## 2.10. Gas chromatography

Gas chromatography (GC) is an analytical technique that is widely used for quality control and additional detection and quantitation of components in a mixture, GC is often used as a tool where the identification of very small amounts is taken into account. It is a common method of chromatography used to segregate and investigate the gas or liquid that can be vaporized without disintegration as a part of analytical science [60].

## 2.11. Metal oxide semiconductor-based gas sensors

Over the past decades, different types of sensors have been developed by using various sensing materials and different transduction stages [61]. MOS-based gas sensors stand apart from others because of their easy underlying mechanism and lower cost [62]. Various hazardous gases such as CO, $CO_2$, $NH_3$, $N_2O$, $SO_2$, $O_3$, LPG, $CH_4$, etc. are detected successfully and efficiently by using MOS-based sensors. In general, MOSs such as $SnO_2$, ZnO, $In_2O_3$, $TiO_2$, $CeO_2$, $Fe_2O_3$, $WO_3$, CdO, CuO, etc. are used for the detection of GHGs. There are numerous techniques to synthesize NMOSs for GHGs sensors such as chemical vapour deposition, hydrothermal, co-precipitation, sol-gel method, pulsed laser deposition (PLD), radio frequency (RF) and direct current (DC) sputtering, thermal evaporation, etc. [63,64].

The essential materials used as a gas detector incorporate MOSs, characteristic conduction polymers, conductive composites polymers, metal oxide/composite polymers and other new materials. These materials can be used on various transduction units, also called chemiresistive SAWs, QCM, optical transducer and MOS field-effect transistor (MOSFET). Based on observations, the chemiresistive MOS has great potential to be used for novel gas sensors [50]. The chemiresistor is the easiest and smallest in the size estimation method with a straightforward detection mechanism. There are two primary sorts of MOS-based sensors including n-type whose larger part charge bearer is an electron (such as tin dioxide ($SnO_2$), titanium dioxide ($TiO_2$), zinc oxide (ZnO), iron oxide ($In_2O_3$), etc.), and p-type whose majority charge transporter is hole (such as cobalt oxide (CoO), nickel oxide (NiO), etc.) [65]. Potential uses of these chemical sensors incorporate natural inspections, and aviation vehicle wellbeing observation and several others [66].

Chemiresistive (MOS-based) gas sensors show change in resistance upon exposure to the gases by the oxidative interactions with the negatively charged chemisorbed oxygen [64]. The gas sensing mechanisms such as, gas reaction, reaction rate and selectivity are significantly affected by the surface nature, porosity, microstructure of the detecting material, the existence of catalysts and the sensing temperature [67]. Generally, the operating temperature and thickness of the film affects the response of MOS-based sensors. The response to a specific gas can be incredibly boosted by including a catalytic metal to the oxide layer, however, unnecessary stacking can decrease the response [68]. The sensitivity and selectivity to the specific gases are also affected by the grain size of the oxides because grain boundaries act as an electron scattering centre [69]. Different types of gas sensors are shown in figure 4 and table 2 briefs the transduction mechanism and principle of the various gas sensors.

### 2.11.1. Gas sensing mechanism

The change in electrical conductivity or resistivity of MOSs is the basic principle of gas detection in MOS-based gas sensors [71,72]. In MOS, under the normal atmospheric conditions and typical operating temperature, an electron-depleted surface layer is developed in the presence of atmospheric oxygen that is adsorbed or chemisorbed on the surface. At first, oxygen is consumed by the metal oxide surface when the surface layer is exposed to air, i.e. the oxygen ionic species $O_2^-$, $O^-$ and $O^2$ get adsorbed on the top of the metal oxide grains. This will prompt a band twisting, and a depletion region called the space charge field is formed. When the target gas particles arrive at the surface of the metal oxide grains, they will interact with the oxygen anions and change the concentration level

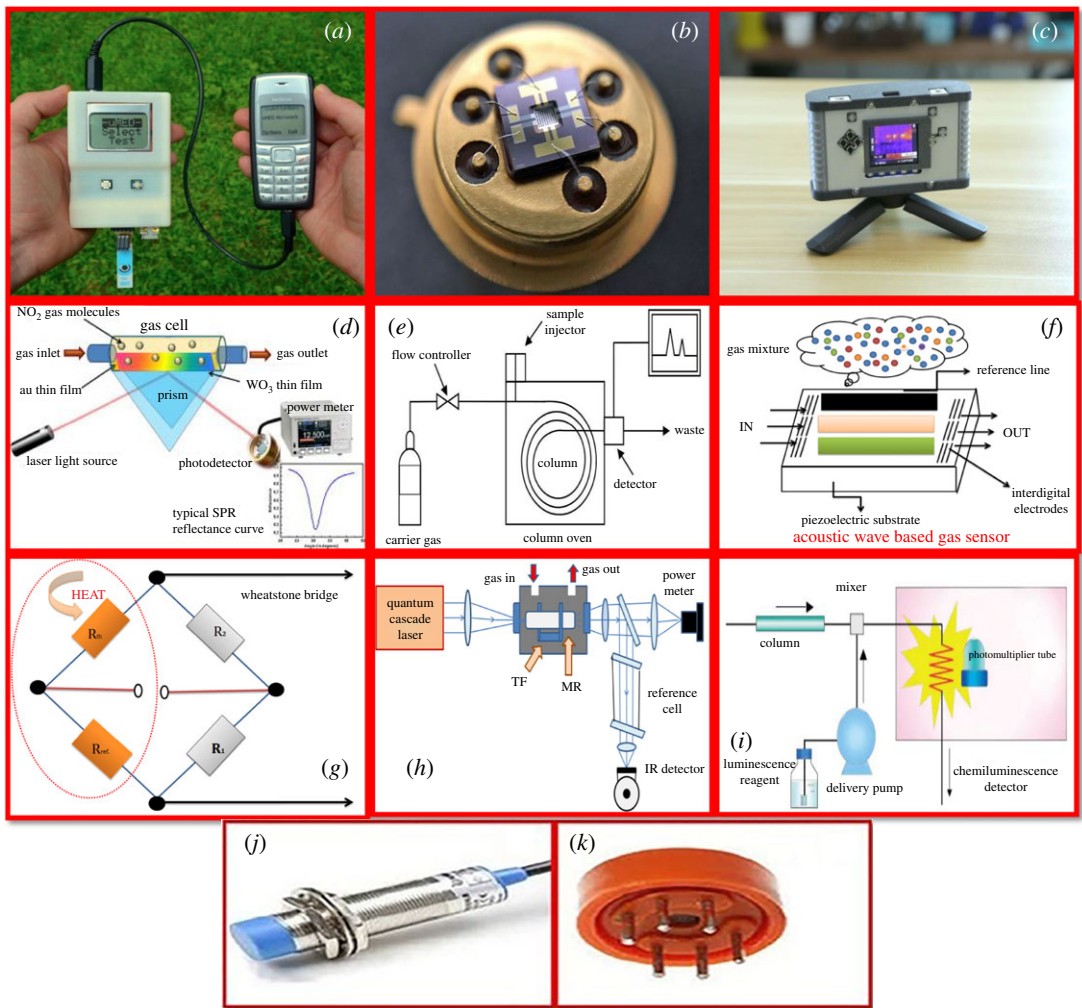

**Figure 4.** (*a*) Electrochemical sensor in the form of device [67], (*b*) MOS sensor [68], (*c*) thermal sensor, in the form of camera [69], (*d*) optical sensor [70], (*e*) gas chromatography, (*f*) mass sensitive sensor, (*g*) catalytic sensor, (*h*) photoacoustic spectroscopy, (*i*) chemiluminescence, (*j*) magnetic sensor, and (*k*) chemical sensor.

of electrons in the metal oxide substances. This will result in a change in conductivity, thereby providing an electronic response signal that can be measured. The detection mechanism in metal oxide gas sensors is identified with ion sorption of species over their surfaces. At the point when the gas sensor is exposed to oxygen, the adsorbed oxygen particle will be framed with the oxygen atoms removing the electrons from the metal oxide inside. The following reaction ((2.1)–(2.4)) steps show the adsorption kinematics [73–75]:

$$O_2(gas) \Leftrightarrow O_2 \text{ (absorbed)}, \tag{2.1}$$

$$O_2(absorbed) + e^- \Leftrightarrow O_2^-, \ (<100°C), \tag{2.2}$$

$$O_2^- + e^- \Leftrightarrow 2O^- \ (100 - 300°C), \tag{2.3}$$

and

$$O^- + e^- \Leftrightarrow O^{2-} (>300°C). \tag{2.4}$$

The types of chemisorbed oxygen ions are determined by the operating temperature of gas sensors. For temperatures below 100°C, between 100°C and 300°C and more than 300°C, the oxygen ions are $O_2^-$, $O^-$ and $O^{2-}$, respectively [76].

Naturally unsafe gases can be segregated into two classes depending on the oxidizing and reducing impacts. $NO_2$, $NO$, $N_2O$ and $CO_2$ gases fall into the category of oxidizing while $H_2S$, $CO$, $NH_3$, $CH_4$ and $SO_2$ gases in reducing. When an n-type MOSs gas sensor is exposed to oxidizing gas, the target gas reacts with the ambient oxygen ions and retains the electrons at the surface. It reduces electron concentration in MOSs. Because electrons are the majority charge transporters in MOSs, the conductance of n-type MOSs decreases on exposure to oxidizing gas. In the case of a p-type MOSs gas sensor, holes are the majority charge transporters. The concentration of holes inside the MOSs is increased owing to the extracted

**Table 2.** Different types of gas sensors and their advantages and disadvantages.

| type of sensor | measured quantities | principle | advantages | disadvantages |
|---|---|---|---|---|
| MOSs-based sensors | conductivity | conductometric | wide range of target gases, fast response, low cost and long lifetime | high-energy consumption, sensitivity to environmental factors, non-selective |
| electrochemical gas sensor | charge, current, voltage, resistance, inductance, etc. | potentiometric, amperometric, resistive, etc. | can measure toxic gases in low concentration | easy contamination |
| magnetic gas sensor | magnetic flux density, magnetic moment, etc. | paramagnetic | consumes low power, relatively affordable | sensitivity to environmental factors |
| thermometric gas sensors | temperature, specific heat, heat flow, etc. | calorimetric | easy to operate in absence of oxygen, low cost and adequate sensitivity for industrial detection | risk of explosion, intrinsic deficiencies in selectivity |
| catalytic gas sensor | temperature, resistance | catalytic/gas oxidation | simple, low cost, measures flammability of gases | requirement of air or oxygen to work |
| chemical gas sensor | composition, concentration, pH, etc. | changes in properties | simple design and low cost | cross sensitivity of other gases, limited temperature range |
| optical gas sensor | light intensity, wavelength, polarization, etc. | fluorescence, optical, etc. | simple operational process in absence of oxygen, unaffected from electromagnetic interference | high cost and difficulty in miniaturization |
| mass resistive gas sensor | change in the characteristics such as amplitude and velocity | acoustic | long lifetime and avoiding secondary pollution | sensitive to environmental change |
| gas chromatography | mobile phase (gas and liquid) | partition co-efficient | high sensitivity and selectivity | high cost, difficulty in miniaturization for portable applications |
| chemiluminescence | photocurrent/dark current | emission of radiation | high sensitivity, quick response | nonlinear behaviour |
| photoacoustic spectroscopy | absorbed electromagnetic energy | photoacoustic effect | high sensitivity | stability, miniaturization, integration and selectivity |

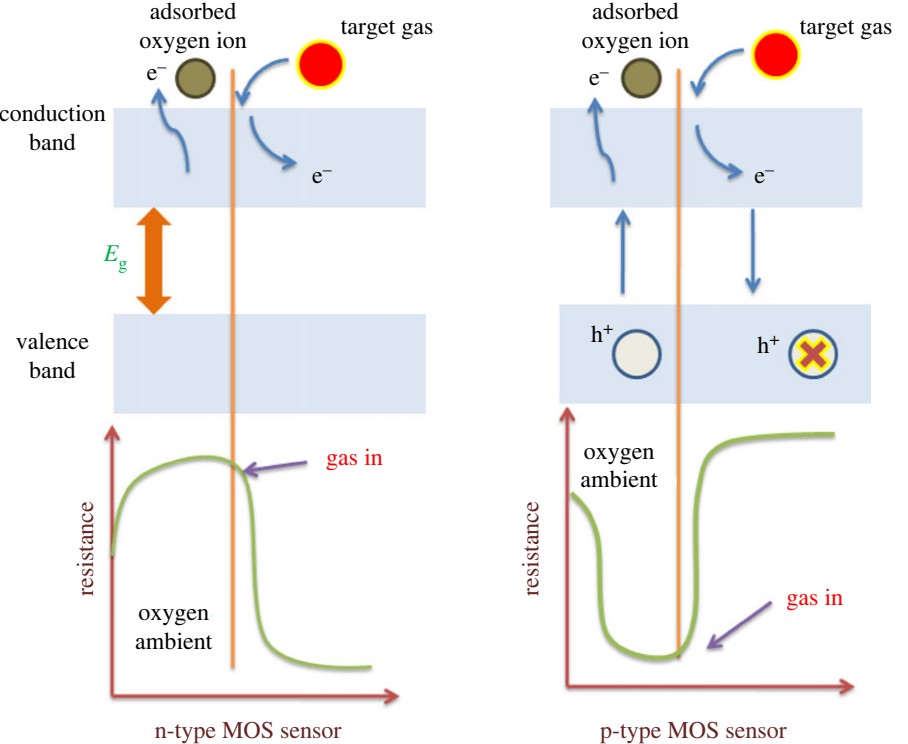

**Figure 5.** Schematic diagram: sensing mechanism of n-type and p-type MOSs.

electron. So, it implies that on the exposure of oxidizing gases, the conductance of p-type MOS increases. The schematic diagram for the sensing mechanism of n-type and p-type MOSs is shown in figure 5.

The following oxidizing reactions (equations (2.5)–(2.8)) show the reaction path between metal oxide and oxidizing gases [74–80]:

$$N_2O \ (gas) + e^- \rightarrow N_2O^-(ads) \tag{2.5}$$

$$N_2O^-(ads) \rightarrow N_2(gas) + O^-(ads) \tag{2.6}$$

$$CO_2(gas) + e^- \rightarrow CO^{2-}(ads) \tag{2.7}$$

and
$$CO^{2-}(ads) + O^-(ads) + 2e^- \rightarrow CO(gas) + 2O^{2-}(ads) \tag{2.8}$$

For n-type MOSs, the gas sensing response can be defined by the following equation:

$$S^n = \frac{R_g}{R_a}, \tag{2.9}$$

while the response of gas for p-type semiconductor oxide to oxidizing gas is generally defined by the following equation:

$$S^p = \frac{R_a}{R_g}. \tag{2.10}$$

Here, $R_g$ and $R_a$ are the electrical resistances of the sensor estimated in the presence of gas and net dry air, respectively. The sign of resistance changes upon the presence/absence of oxidizing and reducing gases for n-type and p-type MOS-based sensors is given in table 3.

## 2.11.2. Gas sensors parameters

### 2.11.2.1. Response
The resistance of the sensor is altered when the sensor is exposed to a gas. The change in resistance of a gas sensor in the air and in the target gas as a function of analyte gas concentration is defined as response. The above equations (2.9) and (2.10) define the response behaviour for n-type MOSs and p-type MOSs, respectively.

**Table 3.** The type of resistance changes upon presence/absence of oxidizing and reducing gases for n-type and p-type MOS-based sensors.

| type of sensitive material | type of target gas | resistance change | response |
|---|---|---|---|
| n-type | oxidizing | resistance increase | $S = R_g/R_a$ |
| n-type | reducing | resistance decrease | $S = R_a/R_g$ |
| p-type | oxidizing | resistance decrease | $S = R_a/R_g$ |
| p-type | reducing | resistance increase | $S = R_g/R_a$ |

### 2.11.2.2. Selectivity

This is the measurement of the ability of the gas sensor to differentiate one gas among the mixture of gases at the same concentration level.

### 2.11.2.3. Sensitivity

This is concerned with the magnitude of change in electrical resistance owing to species of the target gas.

### 2.11.2.4. Limit of detection

Under the given conditions, the capability of the sensor to detect the least possible concentration of the analyte gas is defined as the limit of detection (LOD) of the sensor.

### 2.11.2.5. Working temperature or operating temperature

The temperature at which the gas sensor shows the supreme response for a certain concentration of the analyte gas is known as the working temperature for a gas sensor.

## 3. Greenhosue gas sensors

### 3.1. Metal oxide semiconductor-based carbon dioxide gas sensors

The effect of $CO_2$ on global warming is very large. So it is necessary to detect and control its emission and $CO_2$ sensors can play a significant role to observe and control indoor air quality. These sensors are affordable, very sensitive and can work at room temperature. These sensors can be easily implemented in agricultural applications, the Earth's atmosphere, etc. and are nature friendly and user friendly as well. $CO_2$ sensors play a vital role in the food processing industry as a preservative, in medical care as breath analysers and in biotechnology as environmental incubators [81].

A large number of investigations on metal oxide materials have been reported for $CO_2$ detection, the summary of various MOS-based $CO_2$ gas sensors is given in table 4. The table includes detail of various $CO_2$ sensors based on pure MOSs and metal-doped MOSs, MOS-based composites and hetrojunctions with their synthesis methods and sensor parameters like; operating temperature (°C), response/recovery time ($t_{res}/t_{rec}$) and LOD, etc. The ZnO nanorods with and without adding a catalyst $ZnSn(OH)_6$ (ZHS) microcubes have been synthesized on a p-type silicon substrate by a simple hydrothermal method and it is found that ZHS has a good ability to be used as a catalyst. In this report, ZHS microcubes can enhance the sensing performance of $CO_2$ by 35% in comparison to previously reported studies [104]. Ca-doped ZnO thin films coated Langasite ($La_3Ga_5SiO_{14}$) substrate have been prepared which shows good sensing performance towards $CO_2$ at high temperature by using a SAW sensor. Maximum $CO_2$ sensing is found to be 25 000 ppm with 2.469 kHz response at 400°C [88]. $CO_2$ sensors based on ZnO nanorods of length (1–3 µm) doped with Ge, Nd and W, etc. were synthesized by using the mechanochemical combustion method. The doped ZnO nanorods were found to be more sensitive than the undoped-ZnO under an air atmosphere [94]. Owing to the inclusion of Na doping, the surface of plane ZnO film turned to a wrinkle network having granular structure. This nanostructured 2.5% Na-doped ZnO film showed high sensitivity (81.9%) for $CO_2$ gas in comparison to 1.0% for pure ZnO film. The recovery and response time of Na-doped ZnO was also increased owing to the doping of Na atoms [91]. Multifunctional ZnO thin film shows maximum sensitivity of 400 ppm at 350°C and good response and recovery time of 75 and 180 s, respectively, towards $CO_2$ gas [90]. Organic materials featuring ethynylatedthiourea derivatives have been

**Table 4.** Summary of various MOS-based carbon dioxide ($CO_2$) gas sensors. (C = concentration; $t_{res}/t_{rec}$ = response time/recovery time; LOD = limit of detection; response is defined as $R_a/R_g$ (for reducing gases) or $R_g/R_a$ (for oxidizing gases), $R_a$: resistance of the sensor exposed to air, $R_g$: resistance of the sensor exposed to the target gas.)

| material | structure | synthesis method | target gas | C (ppm) | operating temp. (°C) | response | $t_{res}/t_{rec}$ | LOD | ref. |
|---|---|---|---|---|---|---|---|---|---|
| BiOCl–Au | nanopartides | surfactant-assisted | $CO_2$ | 400 | 300 | 63.2 | 1.3/1.5 s | 100 ppm | [82] |
| Ba/SmCoO₃ | powders | aqueous solution | $CO_2$ | — | 373 | ~1.5 | 202 s | — | [83] |
| Zn/SnO₂ | thin films | spray pyrolysis | $CO_2$ | 500 | 300 | 90 | 55/82 s | — | [84] |
| LaOCl/SnO₂ | nanofibres | electrospinning | $CO_2$ | 1000 | 300 | 3.7 | 24/92 s | 100 ppm | [85] |
| La/ZnO | nanopowder | hydrothermal | $CO_2$ | 5000 | 400 | 65 | 90/38 s | 100 ppm | [86] |
| ZnO/Ca | nanopowders | modified sol-gel | $CO_2$ | 50 000 | 200 | ~9 | — | 2500 ppm | [87] |
| Ca/ZnO | thin film | wet chemical | $CO_2$ | 25 000 | 400 | ~2.5 | 87/132 | 5000 ppm | [88] |
| polyaniline/LaFeO₃ | microsphere | hydrothermal | $CO_2$ | 20 000 | RT | 31.8 | 334.2/86.8 s | 5000 ppm | [89] |
| ZnO | thin film | spray pyrolysis | $CO_2$ | 400 | 350 | 64 | 75/108 s | 25 ppm | [90] |
| ZnO/Na | nanostructured films | spin-coated | $CO_2$ | 50 | RT | 81.9 | 283/472 s | — | [91] |
| Gd/CeO₂ | nano-pellets | co-precipitation | $CO_2$ | 800 | 250 | 45 | — | — | [92] |
| ethynylated-thiourea | solution | reaction | $CO_2$ | 1000 | RT | 25 | 1/3 min | 249 ppm | [93] |
| W/ZnO | nanorods | mechanochemical combustion | $CO_2$ | 1000 | 450 | ~65 | ~15/~20 s | 100 ppm | [94] |
| MWCNT | nanotube | DLICVD | $CO_2$ | 5000 | 30 | 2.1 | 30.2/49.6 s | 1670 ppm | [95] |
| CuO–Cu$_x$Fe$_{3-x}$O₄ | nanocomposite thin film | RF sputtering | $CO_2$ | 5000 | 250 | 0.50 | 9.5/— h | — | [96] |
| Ag–BaTiO₃–CuO | thin films | RF sputtering | $CO_2$ | 5000 | 250 | 0.28 | 15/10 min | 500 ppm | [97] |
| La$_{1-x}$Sr$_x$FeO₃ | nanocrystalline powders | sol–gel | $CO_2$ | 2000 | 380 | 0.25 | 11/15 min | 500 ppm | [98] |
| CdO | nanowires | microwave-assisted wet chemical | $CO_2$ | 5000 | 250 | 0.01 | 3.33/5 min | 2000 ppm | [99] |
| SnO₂/ZnO | composites | screen printing | $CO_2$ | 60 | RT | ~0.79 | — | ~20 ppm | [100] |
| La₂O₂CO₃ | nanopartides | hydrothermal | $CO_2$ | 5000 | 300 | 0.62 | 53/120 s | 300 ppm | [101] |
| TiO₂–PANI | thin film | spin coating | $CO_2$ | 1000 | 30 | 53 | 9.2/5.7 min | — | [102] |
| MoWO₃ | nanostructured thin films | RF magnetron co-sputtering | $CO_2$ | 0.5 | RT | ~29.2 | 6.53/8.05 s | — | [103] |

(*Continued.*)

| material | structure | synthesis method | target gas | C (ppm) | operating temp. (°C) | response | $t_{res}/t_{rec}$ | LOD | ref. |
|---|---|---|---|---|---|---|---|---|---|
| ZnO | nanorods | hydrothermal | $CO_2$ | 1000 | 150 | 0.9 | 11/30 s | 100 ppm | [104] |
| $Al_2O_3$/MWCNT | nanotubes | gel cast | $CO_2$ | 450 | RT | 0.07 | 53.7/— s | 50 ppm | [105] |
| $La_2O_3$/$SnO_2$ | nanofibres | electrospinning | $CO_2$ | 100 | 300 | 5.1 | — | — | [106] |
| ZnO | thin film | magnetron sputtering | $CO_2$ | 1000 | 300 | 1.01 | <20/20 s | 500 ppm | [107] |
| ZnO | nanowires | sol-gel | $CO_2$ | 15 | 200 | 1.04 | 8/40 s | — | [108] |
| $SnO_2$ | nanoparticles | co-precipitation | $CO_2$ | 2000 | 240 | ~1.3 | ~350/4 s | 2000 ppm | [109] |
| $SnO_2$ | nanoparticles | mechanical milling | $CO_2$ | 1000 | 400 | 1.1 | — | — | [110] |
| CdO | nanopowders | co-precipitating | $CO_2$ | 5000 | 250 | 1.03 | 200/300 s | 500 ppm | [111] |
| CuO | porous film | pneumatic spray pyrolysis | $CO_2$ | 100 | RT | 1.04 | 10/6 s | 20 ppm | [112] |
| $YPO_4$ | nanobelts | surfactant-assisted colloidal | $CO_2$ | 200 | 400 | — | 200/136 s | — | [113] |
| $La_2O_3$ | microrods | chemical bath | $CO_2$ | 350 | 250 | ~1.9 | ~50/73 s | 100 ppm | [114] |
| LaOCl | nanopowders | sol-gel | $CO_2$ | 2000 | 260 | 3.40 | — | — | [115] |
| RGO | nanosheets | airbrushing | | 5000 | RT | 1.02 | — | — | [116] |
| $Nd_2O_2CO_3$ | nanoparticles | sol-gel | $CO_2$ | 1000 | 350 | ~4 | — | 300 ppm | [117] |
| $La_2O_2CO_3$ | nanorods | co-precipitation | $CO_2$ | 3000 | 325 | 7.08 | 15/30 min | 100 ppm | [118] |
| CNT | nanotubes | CVD | $CO_2$ | 800 | RT | 1.1 | — | — | [119] |
| $LaFeO_3$ | nanoparticles | sol-gel | $CO_2$ | 2000 | 300 | ~2.2 | 240/480 s | — | [120] |
| $GdCoO_3$ | nanoparticles | solution polymerization | $CO_2$ | — | 400 | 1.1 | 10/5.3 s | — | [121] |
| RGO | nanosheets | hydrogen plasma | $CO_2$ | 769 | RT | 0.13 | —/~4 min | 300 ppm | [122] |
| $Yb_{0.8}Ca_{0.2}FeO_3$ | nanoparticles | sol-gel | $CO_2$ | 5000 | 260 | 2.01 | 24/31 s | 1000 ppm | [123] |
| graphene | nanosheets | mechanical cleavage | $CO_2$ | 100 | RT | ~1.3 | 8/10 s | 10 ppm | [124] |
| CNT | random CNT network | CVD | $CO_2$ | 500 | RT | 1.2 | 385/412 s | 100 ppm | [125] |
| few-layered graphene | nanosheets | electrochemical exfoliation | $CO_2$ | 200 | RT | 3.8 | 11/14 s | 3 ppm | [126] |

(*Continued.*)

**Table 4.** (Continued.)

| material | structure | synthesis method | target gas | C (ppm) | operating temp. (°C) | response | $t_{res}/t_{rec}$ | LOD | ref. |
|---|---|---|---|---|---|---|---|---|---|
| La$_{0.875}$Ca$_{0.125}$FeO$_3$ | nanopartices | sol-gel | CO$_2$ | 1000 | 320 | ~1.7 | — | — | [127] |
| In$_2$Te$_3$ | thin film | flash evaporation | CO$_2$ | 1000 | RT | 1.1 | 0.05/—s | 100 ppm | [128] |
| In$_2$Te$_3$ | thin film | SHI irradiation | CO$_2$ | 1000 | RT | 1.1 | 15–20/—s | — | [129] |
| CNT | nanotubes | CVD | CO$_2$ | 800 | RT | 1.02 | 12/56 s | 50 ppm | [130] |
| LaOCl–SnO$_2$ | porous film | electrostatic spray pyrolysis | CO2 | 2000 | 425 | ~1.4 | — | 400 ppm | [131] |
| CuO–BaTiO$_3$ | thin film | magnetron sputtering | CO$_2$ | 5000 | 300 | ~1.1 | >120/80 s | 500 ppm | [132] |
| CuO–BaTiO$_3$ | thin film | magnetron sputtering | CO$_2$ | 5000 | RT | 3.3 | 300/300 s | 500 ppm | [133] |
| CuO–BaTiO$_3$ | thin film | magnetron sputtering | CO$_2$ | 1000 | 250 | ~1.8 | >90/120 s | 350 ppm | [134] |
| LaFeO$_3$–SnO$_2$ | porous film | mixing | CO$_2$ | 4000 | 250 | 2.7 | <20/—s | — | [135] |
| ZnO$_2$–CuO | thick film | mixing | CO$_2$ | 4000 | 300 | 1.3 | — | 400 ppm | [136] |
| Ca–ZnO | nanopartices | sol-gel | CO$_2$ | 5000 | 450 | 2.1 | — | — | [87] |
| $_{0.4}$SnO$_2$–$_{0.6}$WO$_3$ | nanopartices | mixing | CO$_2$ | 300 | RT | ~1.1 | 127/42 s | 100 ppm | [137] |
| Cr–TiO$_2$ | thin film | magnetron sputtering | CO$_2$ | 10 000 | 55 | ~1.2 | — | — | [138] |
| CuO–Cu$_x$Fe$_{3-x}$O4 | thin film | RF sputtering | CO$_2$ | 5000 | 250 | 1.9 | 9.5/—h | 1000 ppm | [96] |
| BaCO$_3$–Co3O$_4$ | nanopartices | grounding | CO$_2$ | 1000 | 150 | ~1.1 | 192/215 s | 500 ppm | [139] |
| ppy–FeCl$_3$ | porous film | chemical oxidative polymerization | CO$_2$ | 700 | RT | ~1.6 | 210/1560 s | 100 ppm | [140] |
| SnO$_2$–LaOCl | nanopowders | impregnation | CO$_2$ | 2000 | 350 | 1.02 | — | 500 ppm | [141] |
| SnO$_2$/ZIF-67 | core–shell | mixing | CO$_2$ | 5000 | 205 | 1.2 | 220/25 s | — | [142] |
| SnO$_2$–LaOCl | nanowires | drop-coating | CO$_2$ | 4000 | 400 | 6.8 | 15/19 s | 250 ppm | [143] |
| SWCNT/PIL | nanotubes | grinding | CO$_2$ | 10 | RT | 1.02 | <60/—s | 500 ppt | [144] |
| ZnO–LaOCl | nanowires | drop-coating | CO$_2$ | 2000 | 400 | 3.5 | ~15/~17 s | — | [145] |
| SnO$_2$–La | nanopartices | impregnation | CO$_2$ | 500 | 250 | 1.4 | ~20/~75 s | 50 ppm | [146] |
| TiO$_2$/Al$_2$O$_3$ | thin films | ALD | CO$_2$ | 25 | RT | 1.4 | — | 5 ppm | [147] |

(Continued.)

**Table 4.** (*Continued.*)

| material | structure | synthesis method | target gas | C (ppm) | operating temp. (°C) | response | $t_{res}/t_{rec}$ | LOD | ref. |
|---|---|---|---|---|---|---|---|---|---|
| CuO–BaTiO$_3$/Ag | thin film | magnetron sputtering | CO$_2$ | 5000 | 300 | 1.2 | 120/80 s | 500 ppm | [97] |
| CuO/BaTiO$_3$ | spheres decorated leaves | co-precipitating | CO$_2$ | 700 | 120 | 1.2 | 5/18 s | ~51 ppm | [148] |
| RGO/PEI | thin films | airbrushing | CO$_2$ | 3667 | RT | ~1.01 | 14/14 s | 20 ppm | [149] |
| ZnO/Ag–CuO | spheres decorated with leaves | impregnation | CO$_2$ | 1000 | 320 | 1.3 | 76/265 s | 100 ppm | [150] |
| In$_2$O$_3$/CaO | mesoporous | impregnation | CO$_2$ | 2000 | 230 | ~1.8 | — | 300 ppm | [151] |
| La$_2$O$_3$/Pd | porous film | dipping | CO$_2$ | 500 | 250 | 1.4 | 105/145 s | 250 ppm | [152] |
| La$_2$O$_3$/Pd | thin film | dipping | CO$_2$ | 400 | 250 | 2.8 | 80/50 s | 50 ppm | [24] |
| graphene/Sb$_2$O$_3$ | thin film | *in situ* chemical route | CO$_2$ | 50 | RT | ~1.2 | 16/22 s | — | [25] |
| PILs/La$_2$O$_2$CO$_3$ | thin films | drop-casting | CO$_2$ | 2400 | RT | ~1.1 | 300/—s | 150 ppm | [153] |

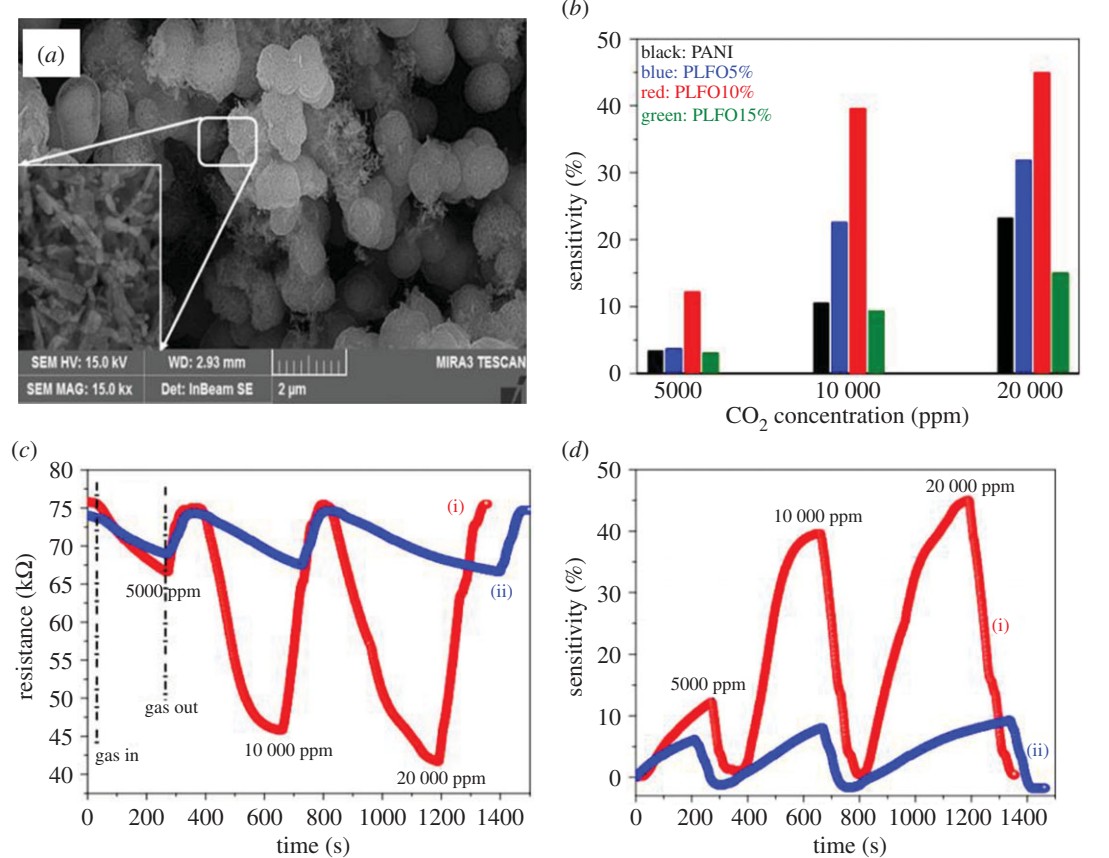

**Figure 6.** (a) Field emission scanning electron microscopy image, (b) sensitivity versus $CO_2$ concentration at room temperature (27°C) and (approx. 90%) humidity of mesoporous $LaFeO_3$ microspheres, (c) resistance shift, and (d) sensitivity of 10% PANI/$LaFeO_3$ film sensor exposed to different concentrations of the $CO_2$ gas (i) at initial test and (ii) after 1 year. Reprinted with permission from [89].

developed to act as active materials for the detection of carbon dioxide. The resistance value of these derivatives was found to increase when it was exposed to $CO_2$ gas [93].

Many researchers reported the preparation of multi-walled carbon nanotubes (MWCNTs)-based $CO_2$ gas sensors using different methods such as the chemical vapour deposition technique followed by the spin coating technique and low-cost gel cast method. The MWCNTs grown on Co nanoparticles exhibited high sensitivity of 2.1 at 5000 ppm, towards $CO_2$ gas, with fast response and recovery time as 30.21 s and 49.62 s, respectively, at room temperature, while CNT composite-based film sensors with different concentrations of alumina ($Al_2O_3$) show excellent changes in conductance when exposed to $CO_2$ ranging from 50 to 450 ppm. The speed and stability of CNT based sensors are found to be favourable for $CO_2$ sensing [95,105]. $SnO_2$-based $CO_2$ sensors have attracted favourable attention of the research community for the detection of different gases owing to effective enhancement in sensitivity.

Recently, many reports have been published on Au-$La_2O_3$-doped $SnO_2$ nanofibres (NFs) (figure 6a), porous $LaFeO_3$/$SnO_2$ nanocomposites and $LaFeO_3$ microspheres fabricated by different chemical methods in which incorporation of metals or metal oxides enhance the response behaviour of the sensor towards $CO_2$ gas. A $LaFeO_3$/$SnO_2$ thick film-based sensor shows low response time (<20 s) at 250°C. While a mesoporous polyaniline (PANI)/$LaFeO_3$ (LFO) nanocomposite (PLFO) gas sensor with 10 wt% LFO shows 13.20 times higher gas sensitivity to 20 000 ppm of $CO_2$ in comparison to a pure PANI gas sensor as shown in figure 6b. These sensors have long-term stability, even after 1 year the response is reasonably good as shown in figure 6c,d. The $La_2O_3$ (5 wt%)-doped $SnO_2$ NFs prepared by an electro spinning process shows effective improvement in the response of many kinds of gases, especially $CO_2$. In addition, Au particles (approx. 15–20 nm) sputtered on 5 wt% $La_2O_3$/$SnO_2$ can improve the response of a gas sensor by 50 [89,106,135]. Copper oxide has also been an attractive material to detect $CO_2$ in different nanodimensions such as thin films, nanocomposites and nanostructures. $CuO$–$Cu_xFe_{3-x}O_4$ nanocomposite (with $0 \leq x \leq 1$) based sensor prepared by RF sputtering was investigated for $CO_2$, concentrations up to 5000 ppm at different operating

temperatures (130–475°C) while $BaTiO_3$–CuO sputtered thin film sensor has been reported to detect $CO_2$ with the addition of Ag. The resistance and capacitance changes are closely related to the changes in the work function in $BaTiO_3$ and CuO material. A similar study was also reported on heterostructures of Ag–CuO/$BaTiO_3$ for low-temperature $CO_2$ detection by the diffuse reflectance infrared Fourier transform spectroscopy (DRIFTS) technique. This metal nanocomposite-based sensor showed significant selectivity and sensitivity towards $CO_2$ at 120°C with remarkable response and recovery times (less than 10 s), high repeatability and accuracy (98%) in comparison to the pure $BaTiO_3$ and CuO. The CuO films deposited by the spray pyrolysis method showed good sensitivity towards different concentrations of $CO_2$ [96,112,132,150].

Table 4 summarizes the materials, synthesis methods and detection parameters of a variety of MOS-based $CO_2$ sensors. Over the years, the sensitivity, selectivity, detection limit and response/recovery time have been remarkably improved with the use of a wide range of materials, synthesis methods and synthesis parameters. It can be inferred that nanostructured composite materials have better sensing performance as compared to the other materials. Numerous reports have proved that the MOS-based sensors have a great potential to be used in commercial $CO_2$ gas sensors.

Various methods have also been used by different researchers for the detection of $CO_2$ like gas chromatography (GC), photoacoustic spectroscopy, etc [154]. In a published report, gas phase $CO_2$ in the headspace of champagne glasses was monitored through combined diode laser spectrometry and micro-GC analysis. It was also discussed in this report that an excess amount of $CO_2$ can even cause a very unpleasant tingling sensation perturbing both ortho- and retronasal olfactory perception [154]. Photoacoustic spectroscopy such as quartz-enhanced photoacoustic spectroscopy (QEPAS) and cantilever-enhanced laser-PAS have been reported for $CO_2$ detection [155]. A report was published by Gerald Gerlach *et al.* [156] which described the various analytical methods like spectroscopy and GC for the detection of $CO_2$.

## 3.2. Metal oxide semiconductor-based methane gas sensors

$CH_4$ is an odourless, colourless but highly flammable GHG which has been widely employed as reliable source of energy for domestic as well as industrial applications. It is an important constituent of natural gas and used as fuel for vehicles (CNG, LNG), heat generation and also in electricity production. It is often generated from biomass combustion, coal mines, sewage treatment, animal waste and rice production, etc. [157,158]. However, $CH_4$ is known for its extremely flammable and volatile nature that sometimes causes dangerous explosions within a concentration range of 4.9–15.4% [159]. Owing to safety concerns, $CH_4$ detection is essential and highly sensitive and reliable sensors are required to prevent deadly explosions. Currently, MOSs-based sensors have gained significant attention for $CH_4$ sensing, the summary of various MOS-based $CH_4$ gas sensors with sensor parameters such as operating temperature (°C), response/recovery time ($t_{res}/t_{rec}$) and LOD, etc. are given in table 5.

Pt-loaded $SnO_2$ NFs with different Pt concentration (10–30 mol%) were prepared using the electrospinning technique followed by calcinations and screen printing [188]. The sensors fabricated with 20 mol% Pt-$SnO_2$ NFs (100–150 nm) show an excellent response time of 4.48 s towards 1000 ppm $CH_4$ at 350°C. Figure 7a shows the schematic diagram of a gas sensor fabrication process, figure 7b,d are transmission electron microscopy (TEM) images, and figure 7c,e are high resolution (HR)-TEM images, that represent the nanoscale microstructure of unloaded $SnO_2$ and Pt-loaded $SnO_2$ NFs, respectively. The HR-TEM image (figure 7e), shows the simultaneous presence of $SnO_2$ and PtO.

Figure 8a,b presents the sensing mechanism of Pt-loaded $SnO_2$ NFs for air and $CH_4$, respectively, while figure 8c shows the sensing response of 20 mol% Pt-$SnO_2$ NFs to 1 ppm–10 ppm $CH_4$ at 350°C which was 2 times less than the minimum detection limit of previously fabricated $SnO_2$-based $CH_4$ gas sensors and (d) shows resistance shift for 20 mol% Pt-loaded $SnO_2$ sensing characteristics towards $CH_4$ at 100°C.

Cr-doped $SnO_2$ nanoparticles with Cr (0–2 wt%) were synthesized by a flame spray technique and sensing film was fabricated using a spin coating technique [173]. The response time of 3.9 s at 350°C towards 1 vol% $CH_4$ offered by 0.5% Cr-doped $SnO_2$ sensor proves that Cr–$SnO_2$ is a promising candidate for selective $CH_4$ detection [173]. Pd-decorated ZnO/rGO hybrid-based $CH_4$ gas sensor shows a fast sensing response time of 74 s at room temperature towards $CH_4$ (25 ppm). This opens a new route to design of ternary hybrid-based $CH_4$ gas sensors [217].

Titanium oxynitride ($TiO_xN_y$) nanostructures prepared by milling showed high sensitivity of 50.12% at room temperature for 20 ppm of $CH_4$. The $\gamma$-$Fe_2O_3$ nanoparticles synthesized by a green approach from leaf extract provides a higher response and selectivity with short response time

**Table 5.** Summary of various MOS-based CH$_4$ gas sensors.

| material | structure | synthesis method | target gas | C (ppm) | operating temp. (°C) | response | $t_{res}/t_{rec}$ | LOD | ref. |
|---|---|---|---|---|---|---|---|---|---|
| TiO$_2$ | nanorods | hydrothermal | CH$_4$ | 60 | RT | 6028 | — | 5 ppm | [160] |
| VO$_2$ | nanorods | thermal evaporation | CH$_4$ | 500 | RT | 35 | 75/158 s | ~100 ppm | [161] |
| Pt/VO$_x$ | thin films | magnetron sputtering | CH$_4$ | 500 | RT | 18.2 | ~16.7/~33 s | ~500 ppm | [162] |
| Au/VO$_2$ | nanosheets | CVD | CH$_4$ | 500 | RT | ~70 | ~50/~100 s | ~100 ppm | [163] |
| Pd/SnO$_2$/rGO | nanoparticles | hydrothermal | CH$_4$ | 4000 | RT | 2.07 | 10 min/— | — | [164] |
| SnO$_2$ | nanoparticles | sol-gel | CH$_4$ | 20 000 | 80 | 74 | 16/70 s | 21 126 ppm | [165] |
| SnO$_2$/WO$_3$ | nanosheets | impregnation | CH$_4$ | 5000 | 90 | 1.5 | ~1.5/~100 s | 5 ppm | [166] |
| γ-Fe$_2$O$_3$ | nanoparticles | green synthesis | CH$_4$ | 100 | 150 | ~8.5 | ~10/~40 s | 1 ppm | [167] |
| SnO$_2$ | quantum dots | sonochemical | CH$_4$ | 5000 | RT | ~10 | ~170/~200 s | — | [168] |
| TiO$_x$N$_y$ | nanopowders | ball milling | CH$_4$ | 100 | RT | 1010 | 33/38 s | 20 ppm | [169] |
| Pd/SnO$_2$ | nanoparticles | sol-gel | CH$_4$ | 937 | 350 | 12.4 | 6/10 s | 47 ppm | [170] |
| Pd/PdO/S-SnO$_2$ | nanoomposites | green recyling | CH$_4$ | 8000 | 240 | 7.8 | 8/12 s | 300 ppm | [171] |
| PANI/polymer/MWCNTs | nanoomposites | wet synthesis | CH$_4$ | 15 | 60 | 3.4 | ~1/—s | 5 ppm | [172] |
| Cr/SnO$_2$ | films | spin coating | CH$_4$ | 250 | 350 | ~1268 | ~3.9/—s | 1 ppm | [173] |
| VO$_2$ | nanoparticles | vapour transport | CH$_4$ | 500 | 150 | 652 | — | 50 ppm | [174] |
| Pd/SnO$_2$ | nanoporous | hydrothermal | CH$_4$ | 3000 | 340 | 17.6 | 3/5 s | — | [175] |
| Pd/SnO$_2$ | hollow spheres | hydrothermal | CH$_4$ | 250 | RT | 4.88 | 3/7 s | — | [176] |
| V$_2$O$_5$ | nanoflowers | magnetron sputtering | CH$_4$ | 500 | 100 | ~8 | 206/247 s | 50 ppm | [177] |
| V$_2$O$_5$ | — | magnetron sputtering | CH$_4$ | 500 | RT | 17 | — | — | [178] |
| VO$_2$ | nanorods | PLD | CH$_4$ | 50 | 50 | ~3.2 | — | — | [179] |
| VO$_x$-MWCNT | nanotubes | CCVD | CH$_4$ | 100 | RT | ~1.5 | 138/234 s | 60 ppm | [180] |
| V$_2$O$_5$ | nanoflakes | RF sputtering | CH$_4$ | 3000 | 330 | 2.8 | ~2.5/~5 min | 50 ppm | [181] |
| ZnO/Zn$_2$SnO$_4$ | microflowers | solvothermal | CH$_4$ | 1000 | 250 | 15.36 | 10/30 s | 400 ppm | [182] |

(Continued.)

**Table 5.** (*Continued.*)

| material | structure | synthesis method | target gas | C (ppm) | operating temp. (°C) | response | $t_{res}/t_{rec}$ | LOD | ref. |
|---|---|---|---|---|---|---|---|---|---|
| SnO$_2$/NiO | porous nanosheets | immersion–calcination | CH$_4$ | 7000 | 330 | 15.2 | 28/44 s | 500 ppm | [183] |
| Pd/SnO$_2$–rGO | nanocomposites | hydrothermal | CH$_4$ | 12 000 | RT | ~9.3 | ~5/7 min | 800 ppm | [164] |
| G-C$_3$N$_4$/ZnO | flower-like/hierarchical | precipitation–calcination | CH$_4$ | 1000 | 320 | ~2.6 | 30/200 s | 100 ppm | [184] |
| SnO$_2$ | nanorods | hydrothermal | CH$_4$ | 10 000 | 150 | 24.9 | 369/350 s | 1000 ppm | [185] |
| NiO/rGO | nanocomposite | hydrothermal | CH$_4$ | 4000 | 260 | 15.2 | 16/20 s | 500 ppm | [186] |
| ZnO/rGO | hybrid composite | hydrothermal | CH$_4$ | 4000 | 190 | 18.5 | 50/60 s | 100 ppm | [187] |
| Pt/SnO$_2$ | nanofibers | electrospinning | CH$_4$ | 1.11 | 350 | 4.5 | 30/150 s | 1 ppm | [188] |
| Fe/SnO$_2$ | thick films | simultaneous precipitation | CH$_4$ | 1000 | 350 | 0.67 | — | 250 ppm | [189] |
| Ca/Pt/SnO$_2$ | thin films | ion beam sputtering | CH$_4$ | 5000 | 400 | 17 | — | 5000 ppm | [190] |
| SnO$_2$ | mesopores | nanocasting | CH$_4$ | 4000 | 600 | 0.6 | ~2/—min | 1000 ppm | [191] |
| Pd/SnO$_2$ | nanopores | surfactant (CTABr) | CH$_4$ | 5000 | 600 | 20 | ~10/~20 s | ~1300 ppm | [192] |
| MoO$_3$ | paste | — | CH$_4$ | 500 | 500 | 10 | ~6/~8 min | — | [193] |
| Pd–Al$_2$O$_3$/SnO$_2$ | catalytic thick film | — | CH$_4$ | 2000 | 450 | ~5 | ~100/—ms | ~2000 ppm | [194] |
| WO$_3$/SnO$_2$ | nanoflowers | impregnation | CH$_4$ | 500 | 110 | ~2.9 | — | 38 ppb | [195] |
| SnO$_2$ | nanosheet | aqueous solution | CH$_4$ | 500 | RT | 1.3 | 18/28 s | — | [196] |
| ZnO/NiO | porous nanosheets | hydrothermal | CH$_4$ | 1000 | 340 | 34.2 | 7/33 s | 300 ppm | [197] |
| Pt/SnO$_2$ | nanocomposites | hydrothermal | CH$_4$ | 500 | 120 | 1.26 | ~4.5/—min | 10 ppm | [198] |
| Al/NiO | thin films | RF sputtering | CH$_4$ | 100 | RT | 58 | 1373/95 s | — | [199] |
| Pd-sensitized ZnO | thin films | ionic layer adsorption reaction | CH$_4$ | 2000 | RT | 2.46 | — | 667 ppm | [200] |
| Pd/ZnO | nanosheets | hydrothermal | CH$_4$ | 5000 | 200 | 19.2 | ~4/~6 min | 100 ppm | [201] |
| ZnO | thin film | electrochemical | CH$_4$ | 100 | 220 | ~4.8 | 24/72 s | — | [202] |
| ZnO | nanowalls | thermal evaporation | CH$_4$ | 100 | 300 | 2 | 6/21 s | 100 ppm | [203] |
| Co/ZnO | microstructure | solvothermal | CH$_4$ | 375 | 140 | 1.05 | 25.2/6.6 s | 100 ppm | [204] |

(*Continued.*)

**Table 5.** (*Continued.*)

| material | structure | synthesis method | target gas | C (ppm) | operating temp. (°C) | response | $t_{res}/t_{rec}$ | LOD | ref. |
|---|---|---|---|---|---|---|---|---|---|
| ZnO/Pd-Ag | nanocrystalline | sol–gel | CH$_4$ | 10 000 | 550 | 31 | ~16.3/— s | 200 ppm | [205] |
| Co/ZnO | microstructure | hydrothermal | CH$_4$ | 100 | 140 | 3.55 | 19/27 s | 50 ppb | [206] |
| ZnO–Ag | ceramics | ceramic technology | CH$_4$ | — | 250 | ~3 | ~40/~60 s | — | [207] |
| ZnO/Pd | nanocomposite | chemical | CH$_4$ | 10 000 | 80 | 36.8 | 7/5 min | 100 ppm | [208] |
| Fe$_3$BO$_6$ | nanoplates | glass liquid | CH$_4$ | 1000 | 252 | 33 | 1.2/2.6 min | 50 ppm | [209] |
| RGO/ZnO | nanoparticles chain-like | anodization and thermal annealing | CH$_4$ | 500 | 450 | 30 | — | 5 ppm | [210] |
| ZnO | microwire | carbothermal reduction | CH$_4$ | 2000 | 400 | 26 | ~12/~28 s | 200 ppm | [211] |
| ZnO/SnO$_2$ | film/nanorods | PECVD | CH$_4$ | 100 | 550 | ~5 | 2/42 s | 50 ppm | [212] |
| Pd/Al$_2$O$_3$ | particles | colloid mixing impregnation | CH$_4$ | 1000 | 400 | — | ~15/~35 s | 1 ppm | [213] |
| Ag/Ag$_2$O–SnO$_2$ | nanocomposites | impregnation | CH$_4$ | 2000 | 170 | 40 | ~5/93 s | 1 ppm | [214] |
| Fe$_3$O$_4$/hydrogel/MWCNTs | nanocomposites | wet synthesis | CH$_4$ | 20 | RT | — | 120/— s | 5 ppm | [215] |
| CdTiO$_3$ | thin films | magnetron co-sputtering | CH$_4$ | 500 | 250 | 3.4 | ~38/~70 s | 100 ppm | [216] |

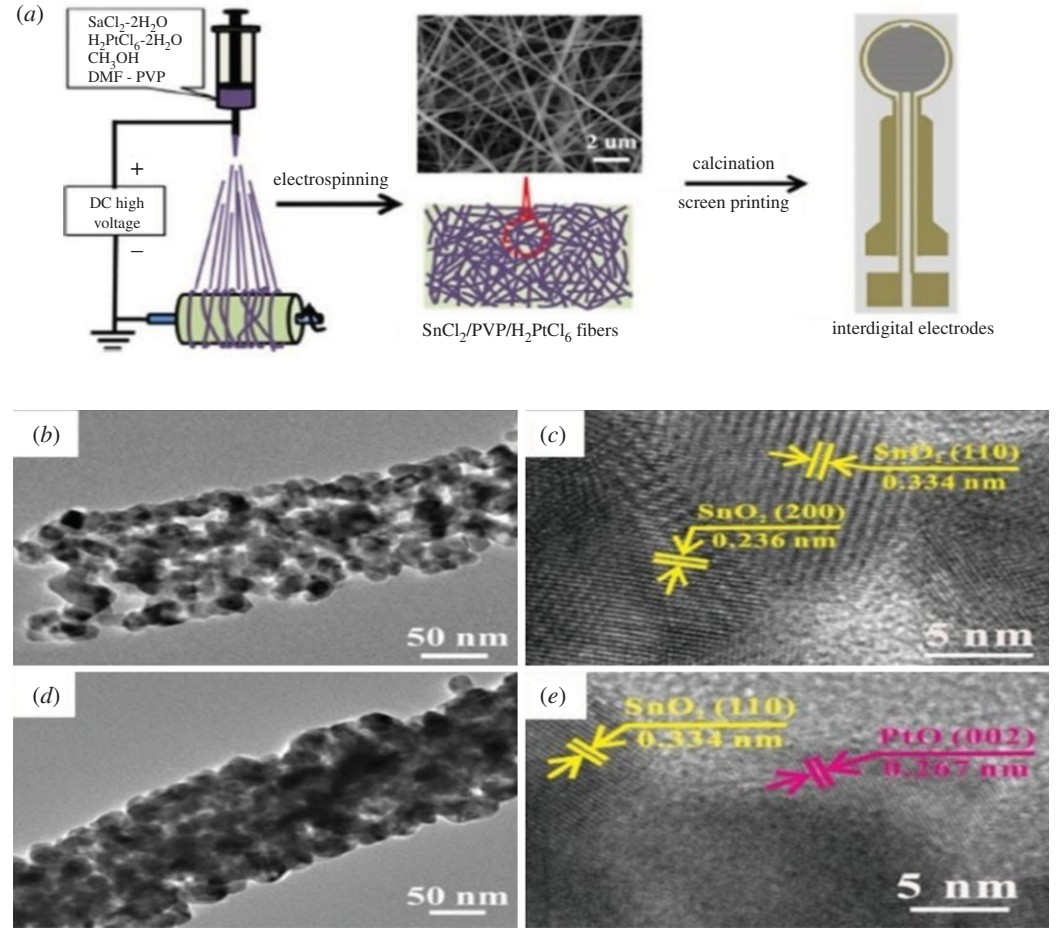

**Figure 7.** (*a*) Schematic diagram showing the process of fabrication of the gas sensor device. (*b*) TEM images of unloaded SnO$_2$, (*c*) HR-TEM images of unloaded SnO$_2$ (*d*) TEM images of 20 mol% Pt-loaded SnO$_2$ NFs, and (*e*) HR-TEM images 20 mol% Pt-loaded SnO$_2$ NFs. Reprinted with permission from [188].

for CH$_4$ sensing than euphorbia extracted γ-Fe$_2$O$_3$ nanoparticles [167]. In RGO-SnO$_2$ heterostructure composite D-glucose and L-ascorbic acid reduced GO showed better CH$_4$ sensing performance at room temperature than sodium borohydrate and hydrazine hydrate reduced GO, while L-ascorbic acid RGO-SnO$_2$ heterostructure showed the highest CH$_4$ sensing response owing to the synergistic effect between dehydroascorbic acid and surface of SnO$_2$ [218]. Thermally evaporated vanadium dioxide (VO$_2$) nanorods showed good sensing response with different concentrations (100–500 ppm) of CH$_4$ at room temperature. The nanorod structure provides a large surface area and sensing sites for CH$_4$ detection [161]. Au-decorated VO$_2$ nanosheets prepared by the CVD method followed by ion sputtering provide good sensing response towards 100–500 ppm of CH$_4$ at room temperature. This is attributed to the formation of a heterojunction at the interface and the creation of a depletion layer within the interface of the Au nanoparticles and VO$_2$ nanosheets [163].

Magnetron-sputtered Pt-loaded VO$_x$ thin films exhibit higher response of 18.2 with 500 ppm concentration of CH$_4$ at room temperature [162]. TiO$_2$ nanorods-based sensors showed excellent response and recovery time of 45 s and 33 s, respectively, with high selectivity towards 199 ppm of CH$_4$ owing to high surface area and point defects of TiO$_2$ [160]. SnO$_2$-loaded WO$_3$ nanosheets showed 1.4 times higher response towards CH$_4$ than pure WO$_3$ at working temperature of 90°C. The highly reactive sites as a result of defect formation at the SnO$_2$-loaded WO$_3$ heterojunction, and oxygen chemisorptions at the dangling bonds of W atoms of WO$_3$ nanosheets result in significant enhancement in the sensing behaviour [166]. The V$_2$O$_5$ nanostructures-based sensor showed a response of about 6.52–8% at 150°C towards 50–500 ppm CH$_4$ concentration and exhibited its specificity towards the C–H bond (CH$_4$) [174,177].

The ZnO–reduced graphene oxide (rGO) nanohybrid composite provides a superior sensing response of 4.52% for sensing CH$_4$ at optimal temperature of 190°C than pure ZnO and rGO. The sensing mechanism is explained on the basis of the formation of a heterojunction of ZnO and rGO [183].

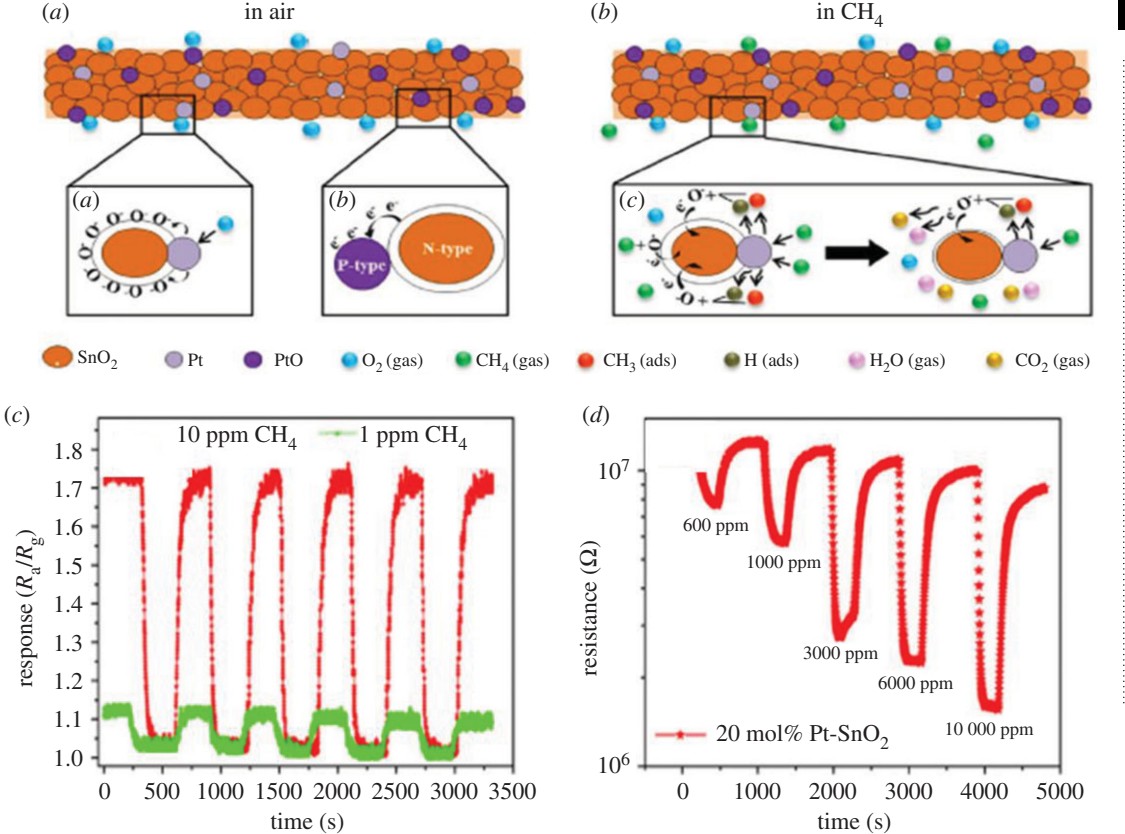

**Figure 8.** Schematic diagram of sensing mechanism of Pt-loaded SnO$_2$ NFs CH$_4$ for: (*a*) air and (*b*) for CH$_4$, (*c*) response of 20 mol% Pt-loaded SnO$_2$ NFs to 1 ppm and 10 ppm CH$_4$ at 350°C, and (*d*) resistance shift for 20 mol% Pt-loaded SnO$_2$ sensing property towards CH$_4$ at 100°C. Reprinted with permission from [188].

Vanadium oxide filled MWCNTs were developed with good response/recovery time of 16 s and 120 s, respectively, at room temperature for CH$_4$ gas. The filling of vanadium oxide in MWCNTs increased the density of states around the Fermi level and adsorption sites [180]. Co-doped ZnO prepared by the hydrothermal process followed by calcinations at 600°C demonstrated superior response time and recovery time of 19 s and 27 s, respectively, towards 100 ppm concentration of CH$_4$, i.e. twice that of the pure ZnO sensor [206]. SnO$_2$ nanorods-nanoporous graphene hybrids with 0.05 weight ratios showed a high sensing response of 24.9–51% at 150°C towards 1000–10 000 ppm of CH$_4$. This was higher than pure SnO$_2$ nanorods owing to the larger surface area of the graphene hybrid and synergistic interaction between nanoporous graphene and SnO$_2$ nanorods [185]. The much better sensitivity (9.5%) for 120 00 ppm CH$_4$ was achieved by a Pd-doped SnO$_2$/rGO composites-based sensor than Pd-doped SnO$_2$/PrGO owing to catalytic action of Pd [164]. Pulsed DC-sputtered nanostructured VO$_2$ thin films showed a reversible semiconductor to metal transition at 60–70°C. The film possesses sensing characteristics at a low temperature of 50°C towards 50 ppm of CH$_4$ [179]. Mesoporous SnO$_2$ of pore size 4.4 nm with large specific surface areas of 80 m$^2$ g$^{-1}$ were prepared by nanocasting mesoporous KIT-silica and it is found to be a highly promising sensing material for CH$_4$ detection [191]. The Pd-modified ZnO nanosheets-based sensor exhibited a remarkable sensing response of 19.20 for CH$_4$ (5000 ppm) with high selectivity and repeatability at 200°C than pure ZnO owing to the spill-over effect and nano-schottky barrier formation [201]. Pd-loaded mesoporous SnO$_2$ hollow spheres based sensors showed response and recovery time of 3 s and 7 s, respectively, with excellent stability of 15 weeks for 250 ppm CH$_4$ compared to the response shown by SnO$_2$ hollow spheres [176]. The NiO nanoparticles decorated in ZnO porous nanosheets increased the specific surface area and interfacial interaction of NiO and ZnO by p–n junction formation. The NiO-decorated ZnO nanosheets exhibited excellent CH$_4$ sensing performance, long-term stability, high response as compared to that of pure ZnO porous nanosheets at 340°C [197].

The research outcomes from various reports in the literature on MOS-based CH$_4$ gas sensors have been summarized in table 5. Nanostructures with specific morphology such as nanorods, nanoflakes,

nanofibres and nanosheets, etc., are known to exhibit better sensitivity, selectivity and response as compared to the conventional materials which is attributed to the larger surface area exhibited by them. Catalytic layers such as Pd or Au significantly improve the sensing properties of the oxide materials. Novel materials such as MWCNTs, reduced graphene oxide and their composites with metal oxides show remarkable sensing behaviour towards $CH_4$.

In addition to MOS sensors other sensors are also available for $CH_4$ detection such as optical sensors [219], calorimetric sensors [220], pyroelectric sensors [221], electrochemical [222] and photoacoustic detection [223]. Li et al. [224] reported $CH_4$ detection based on QEPAS using a high-power continuous wave, single-mode diode laser with an emission wavelength at 2.3 µm and demonstrated the minimum detection limit of a $CH_4$-QEPAS sensor is 7.9 ppm. A compact and portable photoacoustic gas sensor was developed for CH4 detection at 1.6 µm by a software-based wavelength stabilization scheme and the CH4 sensor achieved a minimum detection limit of 11.5 ppm at 10 s response time in the concentration range of 400–6300 ppm [225]. Zheng et al. [226] developed a mid-IR methane sensor using a continuous-wave inter-band cascade laser and showed that the sensor functions normally at 1.0–2.1 ppm as the pressure changed from 25 to 800 Torr. Park et al. [227] fabricated a calorimetric sensor with a dual-catalyst structure and successfully detected $CH_4$ between 200 and 2000 ppm at temperatures of 100–400°C.

## 3.3. Metal oxide semiconductor-based nitrous oxide gas sensors

$N_2O$ also known as laughing gas is one of the most important GHGs which causes $O_3$ layer depletion and contributes to global warming and climate change. $N_2O$ is also used in medical practice as an anaesthetic. It is a colourless gas having a sweet odour and produced from the breakdown of nitrogen-based fertilizers while naturally produced from oceans [22]. There are several other anthropogenic $N_2O$ sources such as wastewater treatment, fossil fuel combustion and industrial nylon production. It causes about 300 times more atmosphere warming per unit weight than $CO_2$ and gives rise to the greenhouse effect [27]. After reaching the upper atmosphere, $N_2O$ molecules can stay there for 100 years. It is one of the reasons of $O_3$ layer depletion, therefore, it is necessary to detect it and decompose it into $N_2$ and $O_2$ before release. Thus, there is huge need for the development of $N_2O$ gas sensors in order to protect the environment and human health. Although, there are very few reports are available on the detection of $N_2O$ using MOS-based sensors, some studies are listed in table 6.

Au-loaded tin oxide thin films grown by ArF excimer laser-induced (MOCVD) exhibited good response of 11.5 for 100 ppm $N_2O$ at 210°C [229]. The (0.5 wt%) $Sm_2O_3$ loaded $SnO_2$ exhibited high sensitivity about 1.5 times higher than pure $SnO_2$ towards $N_2O$ at 475°C and allowed the detection of 35 ppm $N_2O$ in air [232]. Kanazawa et al. [231], reported $SnO_2$-loaded with 0.5 wt% SrO showed 3 times higher response than unloaded $SnO_2$ towards $N_2O$ of 10–300 ppm at 500°C. A schematic of an $SnO_2$-based sensor is shown in figure 9a. The sensitivity versus temperature with various concentrations of $N_2O$ is shown in figure 9b–d. It has been found that (0.5 wt%) SrO-loaded $SnO_2$ is found to be more sensitive towards $N_2O$ than pure $SnO_2$. Rout et al. [233] investigated the sensing characteristics of ZnO, $In_2O_3$ and $WO_3$ nanowires for $N_2O$ detection and found that $In_2O_3$ nanowires of approximately 20 nm exhibited sensitivity of 60 for 10 ppm having response and recovery time of about 20 s at 150°C. Also, the $WO_3$ nanowires prepared by the solvothermal method showed sensitivity of about 25 for 10 ppm having response and recovery time of 10 s and 60 s, respectively, at 250°C. The sensitivity of $In_2O_3$ and $WO_3$ nanowires are unaffected up to 90% of relative humidity. Deb et al. [234] reported the $WO_3$ nanowire mats and nanoparticle films were deposited by the hot filament chemical vapour deposition (HF-CVD) method. The response of tungsten trioxide nanowire (mat-like, nanowire networks) and nanoparticle thin films was in the temperature range of 373–773 K. The mat-like nanowire network exhibited higher resistivity change response in comparison to nanoparticle films at temperatures above 523 K. The nanowire mats showed the high sensitivity with much improved response and recovery times of 75 s and 6 min, respectively, at 723 K. The calculated activation energy from the time constant–temperature plot was about 26 kcal mol$^{-1}$ for the nanowire device.

However, there are several other methods available for the detection of $N_2O$ such as chromatography [235], optical methods [236], laser absorption spectroscopy [237], photoacoustic spectroscopy [238] and amperometric microsensor [239]. Ryu et al. [240] used a GS-electron capture detector (GC-ECD) for the determination for $N_2O$ of different concentrations in ancient-air-trapped ice cores. The quantum cascade laser absorption spectrometer, called 'QCLAS' was developed by Mappe et al. [241] in order to monitor in situ GHGs like $N_2O$ and $CH_4$ at high temporal resolution with a high accuracy. Kang et al. [242] evaluated the repeatability of photoacoustic spectroscopy and found it to be 1.12%,

**Table 6.** Summary of various MOS-based $N_2O$ gas sensors.

| material | structure | synthesis method | target gas | C (ppm) | operating temp. (°C) | response | $t_{res}/t_{rec}$ (s) | LOD | ref. |
|---|---|---|---|---|---|---|---|---|---|
| $Mg_{0.5}Zn_{0.5}Fe_2O_4$ | nanopowder | wet chemical route | $N_2O$ | 1600 | 300 | 19% | — | — | [228] |
| $Au/SnO_x$ | thin films | chemical vapour deposition (CVD) | $N_2O$ | 100 | 210 | 11.5 | — | — | [229] |
| $SnO_2$ | thick films | screen printing | $N_2O$ | 100 | RT | 0.58 | — | — | [230] |
| $WO_3$ | powder | co-precipitation | $N_2O$ | 300 | 450 | 1.32 | — | — | [231] |
| $SnO_2$ | powder | co-precipitation | $N_2O$ | 300 | 450 | 1.66 | — | — | [231] |
| $ZnO$ | powder | co-precipitation | $N_2O$ | 300 | 450 | 1.21 | — | — | [231] |
| (Sr, Ca, Ba, Bi, Sm) loaded $SnO_2$ | powder | co-precipitation | $N_2O$ | 300 | 500 | 4.3 | — | — | [231] |
| $Sm_2O_3/SnO_2$ | powder | electrochemical method | $N_2O$ | 100 | 450 | — | 90/18 | 35 | [232] |
| $In_2O_3$ | nanowires | anodic alumina membrane (AAM) | $N_2O$ | 10 | 150 | 60 | 20/20 s | — | [233] |
| $WO_3$ | nanowire | solvothermal method | $N_2O$ | 10 | 250 | 25 | 10/60 s | — | [233] |
| $WO_3$ | mat-like networked nanowire | HF-CVD | $N_2O$ | 1 | 723 K | — | 75 s/6 min | 100 ppb | [234] |

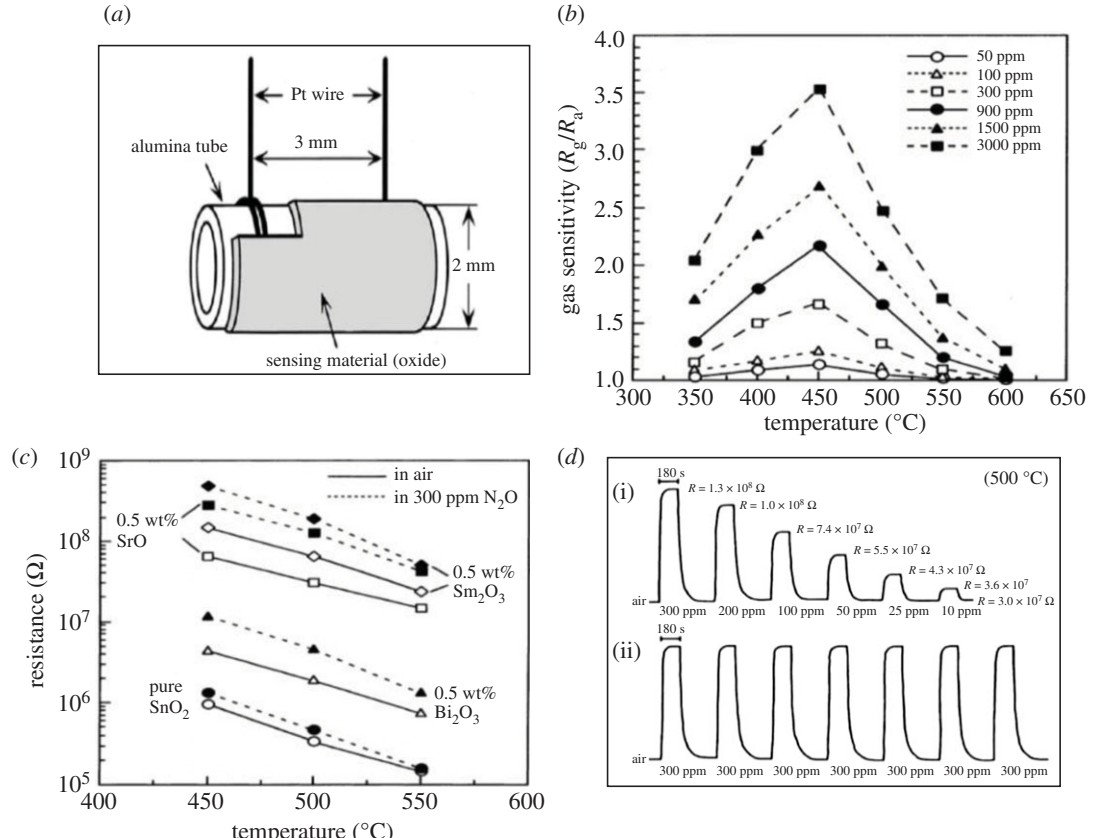

**Figure 9.** (*a*) Schematic diagram of N$_2$O sensor element, (*b*) dependence of sensitivity on temperature at 50–300 ppm for pure SnO$_2$, (*c*) dependence of resistance on temperature for loaded SnO$_2$, and (*d*) (i and ii) response transients to N$_2$O at 500 C for (0.5 wt %) SrO-loaded SnO$_2$.Reprinted with permission from [231].

which is less than the repeatability of 3.0% according to the International Organization for Standardization (ISO) 1564 standard. The detection limit was obtained at 0.025 ppm and the response time was found to be 3 min and 26 s. The major advantage of photoacoustic detection is that there is no need of any complex instrumentation to work at atmospheric pressure. GC-ECD is the most widely employed analytical method for measuring N$_2$O. It is a low-cost method as compared to other analytical techniques [243]. However, optical techniques such as Fourier-transform infrared spectroscopy (FTIR) and laser absorption spectroscopy have major advantages over chromatographic techniques as they are capable to carry out continuous measurements. Laser absorption spectroscopy allows rapid as well as highly sensitive measurements and have lower interference of other trace gases but it is an expensive technique and also cryogenic cooling is a major requirement of this technique [244].

Electrochemical amperometric sensors have the main advantages of gas detection because of their good sensitivity, simplicity, ease of use and low cost of device construction [245]. Some new methods like amperometric biosensors [239] and optical fibres [246] are still developing for N$_2$O detection. There are a number of analytical techniques other than MOS-based sensors for the detection of N$_2$O and each technique has its own advantage and disadvantage. From past years it is possible to measure very low concentrations of N$_2$O very fast. The major requirement in N$_2$O detection is a cheap, accurate and easily deployable sensor which can collect data over extensive areas.

Thus, the outcomes from above-mentioned reports on MOS-based N$_2$O gas sensors suggests that different morphologies such as nanowire, nanopowder, mat-like networked nanowire exhibited better sensitivity and response as compared to conventional materials owing to a large surface area acquired by them. Additionally, SnO$_2$ and WO$_3$ materials have been studied widely for the N$_2$O detection while In$_2$O$_3$ also shows remarkable sensing behaviour towards N$_2$O.

## 3.4. Metal oxide semiconductor-based ozone gas sensors

The peak concentrations of O$_3$ are now substantially higher than the preindustrial levels, owing to many sources of pollution [247,248]. In the Earth's lower (near ground) atmosphere, O$_3$ is formed when

pollutants emitted by cars, power plants, industrial boilers, refineries, chemical plants and other sources react chemically in the presence of sunlight. $O_3$ pollution naturally appears as a concern during the summer, when the weather conditions form ground-level $O_3$ owing to the high temperatures. $O_3$ acts as a GHG, absorbing some of the IR energy radiated by the Earth's surface. Measuring the quantity of the GHG potency of $O_3$ is not easy because its concentration is not uniform across the globe. However, the most widely accepted scientific analysis relating to climate change/global warming suggests that the radiation absorption by atmospheric $O_3$ is about 25% that of $CO_2$ [249]. The environmental $O_3$ does not show any strong global effects owing to its short life but $O_3$ in the atmosphere has a radiative forcing effect approximately 1000 times as strong as $CO_2$ [250].

The demand for $O_3$ sensors is increasing gradually for monitoring and controlling gas emission in the atmosphere. The MOS-based gas sensing device has drawn the interest of many researchers to improve the sensing performance in terms of its selectivity, sensitivity, operating temperature, response and recovery time [251–255]. Various NMOS-based $O_3$ gas sensors, such as ZnO [255–263], $SnO_2$ [264–266], $WO_3$ [267–269], CuO [270], $In_2O_3$ [271–276] and $ZnO/SnO_2$ [277], etc., with their fabrication method and sensor parameters such as operating temperature (°C), selectivity, stability, response/recovery time ($t_{res}/t_{rec}$) and LOD have been demonstrated in table 7. The sensing mechanism of NMOS for $O_3$ is based on the adsorption of gaseous molecules (oxygen, of air) on the surface of the metal oxide. This oxygen ($O_2$) molecule becomes partially ionized with the core electrons to form oxygen species on the surface of the metal oxide: $O_{2s}^-$, $O_s^-$ or $O_s^{2-}$ [290,291]. When $O_3$ is introduced, $O_3$ molecules react with oxygen species on the surface of sensing materials, forming oxygen gas ($O_2$), in addition to releasing the trapped electron back to the core. During this process, the resistance of sensing materials is increased with an increase in $O_3$ gas concentration, based on the following reactions [280,292]:

$$(O_2)_{gas} + (O_2)_{ads}, \tag{3.1}$$

$$(O_2)_{ads} + e^- (O_2^-)_s \tag{3.2}$$

$$(O_2^-)_s + e^- (2O^-)_s \tag{3.3}$$

$$(O^-)_s + (O_3)_{gas}(2O_2)_{gas} + e^-. \tag{3.4}$$

It is well known that the gas sensing mechanism of $O_3$ involves the adsorption of $O_3$ gas molecules onto the surface of sensing materials. This creates an electron-depletion layer which is owing to the adsorption of ions. Thus, this electron-depletion layer increases the potential barrier and consequently increases in resistance of sensing materials [259]. One of the remarkable features of NMOS-based $O_3$ gas sensors is their fast response owing to the high reduction rate of $O_3$ molecules.

A spray pyrolysis deposited $SnO_2$ thin film sensor exhibited a very fast response time of 5–10 s towards 4–1 ppm $O_3$ at 200°C [264], while $SnO_2$-based thin sensitive double layers (approx. 100 nm) demonstrated a response value (approx. 1.2) and response/recovery times (2/3 min) towards 217 ppm $O_3$ at room temperature with LOD of 58 ppm.

One-dimensional ZnO nanorod-like structures prepared by the hydrothermal method showed response/recovery times of 14/60 s towards 0.1 ppm $O_3$ with long-term stability (six months) at 250°C [256]. A high-density porous zinc oxide (ZnO) nanosheets (NSs)-based microelectromechanical systems (MEMS) gas sensor was designed for $O_3$ detection [259]. The sensor demonstrated high response (90.5) towards 100 ppb $O_3$ at an operating temperature of 300°C with long-term stability and repeatability, as it was tested for 168 h. In addition, a ZnO nanosheets-based sensor provides excellent selectivity towards $O_3$ compared to 100 ppb carbon monoxide, methanol, acetone, ethanol as shown [259]. The sensing response of the porous ZnO NSs is attributed to the change in concentration of electrons in the conduction band upon $O_3$ gas exposure, as a result conduction band or acceptor level holes move to the high-energy surface sites residing in the band gap [259].

ZnO multi-wires (50 nm) prepared by the microwave-assisted hydrothermal method provided fast response/recovery times of 9.6/45.6 s with a very high sensitivity to 100 ppb $O_3$ gas at an operating temperature of 120°C [262].

Other NMOSs such as $WO_3$, CuO and PdO have also been reported as good sensing materials for $O_3$ gas sensing [269]. RF magnetron-sputtered $WO_3$ nanostructured thin film exhibited fast response/recovery time value of 1/<60 s with a low LOD (30 ppb) at a temperature of 523 K [269] while sputtered deposited CuO thin film showed a response time of 60 s and very long recovery time of 15 min for 500 ppb $O_3$ at 250°C [270].

**Table 7.** Summary of various MOS-based $O_3$ gas sensors.

| material | structure | synthesis method | target gas | C (ppm) | operating temp. (°C) | response | $t_{res}/t_{rec}$ | LOD | ref. |
|---|---|---|---|---|---|---|---|---|---|
| ZnO | nanorods | hydrothermal | $O_3$ | 0.1 | 250 | ~3 | 14/60 s | 0.06 ppm | [256] |
| ZnO | urchin-like nanorods | CVD | $O_3$ | 280 ppb | 200 | ~100 | — | 280 ppb | [257] |
| ZnO | nanostructures | aqueous chemical | $O_3$ | 1 | RT | ~3.9 | ~3/~5 min | ~1 ppm | [258] |
| ZnO | nanosheets | hydrothermal | $O_3$ | 100 ppb | 300 | 90.5 | — | — | [259] |
| ZnO | powders | polymeric precursor | $O_3$ | 80 ppb | 250 | 5.0 | 11/14 s | 33 ppb | [255] |
| ZnO | thin films | RF magnetron sputtering | $O_3$ | 49.9 | RT | 15 | ~10/~30 min | 0.32 ppm | [260] |
| ZnO | nanowire | ALD | $O_3$ | 600 ppb | 25 | 1.2 | — | 100 ppb | [261] |
| ZnO | nanostructures, flower-like shape | microwave-assisted hydrothermal | $O_3$ | 100 ppb | 120 | ~12 | 9.6/45.6 s | — | [262] |
| ZnO | films | spray pyrolysis | $O_3$ | 182 ppb | RT | 23 | 1/10 min | 16 ppb | [263] |
| $SnO_2$ | thin films | spray pyrolysis | $O_3$ | ~1 | 200 | 4 | 5/2 s | 100 ppb | [264] |
| $SnO_2$ | thin films | SILD | $O_3$ | ~1 | 200 | ~100 | 4/100 s | 1000 ppb | [265] |
| $SnO_2$ | thin films | sol–gel | $O_3$ | 0.5 | RT | 3.1 | 15/12 min | — | [266] |
| $SnO_2$ | thin films | sol–gel | $O_3$ | 217 ppb | RT | ~1.2 | 2/3 min | 58 ppb | [278] |
| $WO_3$ | thin films | — | $O_3$ | 68 ppb | 500 | ~360 | ~300/—s | 13 ppb | [267] |
| $WO_3$ | thin films | RF sputtering | $O_3$ | 80 ppb | 400 | 5 | — | 10 ppb | [268] |
| $WO_3$ | thin films | RF-magnetron sputtering | $O_3$ | 0.8 | 250 | 16 | 1/<60 s | 0.03 ppm | [269] |
| CuO | thin films | RF sputtering | $O_3$ | 500 ppb | 250 | — | ~1/~15 min | — | [270] |
| PdO | thin films | thermal sublimation | $O_3$ | 100 ppb | 175 | ~1700 | — | 10 ppb | [279] |
| $In_2O_3$ | thin films | spray pyrolysis | $O_3$ | ~1 | 250 | ~100 | ~10/180 s | 1000 ppb | [271] |
| $In_2O_3$ | thin films | sol–gel | $O_3$ | 400 ppb | 100 | 20 | — | 200 ppb | [272] |
| $In_2O_3$ | nanoparticles | MOCVD | $O_3$ | 60 ppb | RT (UV) | ~4 | — | 10 ppb | [273] |
| $In_2O_3$ | nanoporous particles | nanocasting | $O_3$ | 0.22 | RT, UV assisted | 200 | 2.5/5.3 min | 50 ppb | [274] |
| $In_2O_3$ | urchin-like microspheres | solvothermal | $O_3$ | 40 ppb | 150 | 21.5 | 60/40 s | 10 ppb | [275] |

(Continued.)

**Table 7.** (*Continued.*)

| material | structure | synthesis method | target gas | C (ppm) | operating temp. (°C) | response | $t_{res}/t_{rec}$ | LOD | ref. |
|---|---|---|---|---|---|---|---|---|---|
| In$_2$O$_3$ | nanoparticles | hydrothermal | O$_3$ | 60 ppb | RT with UV | ~1.62 | — | 10 ppb | [276] |
| Au/TiO$_2$ | core–shells | sol–gel | O$_3$ | 0.5 | RT | 1.15 | 2/5 s | 0.4 ppm | [280] |
| Co/SnO$_2$ | thin films | spray pyrolysis | O$_3$ | 1 | 270 | ~10 | — | 1000 ppb | [281] |
| IN2O/SiN$_x$ | films | RF sputtering and PECVD | O$_3$ | 40 ppb | 195 | ~171 | 3/7.5 min | 20 ppb | [282] |
| SnO$_2$/SWNTs | thin films | sol–gel | O$_3$ | 1 ppb | RT | ~0.88 | — | 20 ppb | [283] |
| WO$_3$/rGO | nanocomposites | liquid flame spray | O$_3$ | 10 ppm | 150 | ~370 | 17.1/32.7 s | 0.5 ppm | [284] |
| ZnO/SnO$_2$ | heterojunctions | hydrothermal | O$_3$ | 0.3 | RT | 37.5 | 13/90 s | 20 ppb | [277] |
| ZnCo$_2$O$_4$ | microspheres | co-precipitation | O$_3$ | 560 ppb | 200 | 0.23 | 8/10 s | 80 ppb | [285] |
| Pt/TiO$_2$–SnO$_2$ | nanomaterial | dip coating | O$_3$ | 2.5 | RT (UV) | ~250 | 160/50 s | 500 ppb | [286] |
| Zn$_{0.95}$Co$_{0.05}$O | thin film | spray pyrolysis | O$_3$ | 1040 ppb | 250 | 0.4 | 46/62 s | 20 ppb | [287] |
| Zn$_{0.95}$Co$_{0.05}$O | thin films | polymeric precursors | O$_3$ | 0.89 | 200 | ~3 | 46/360 s | 42 ppb | [288] |
| SrTi$_{0.85}$Fe$_{0.15}$O$_3$ | thin films | electron beam deposition | O$_3$ | 0.8 | 260 | ~3 | 26/72 s | 0.1 ppm | [251] |
| SrTi$_{1-x}$Fe$_x$O$_3$ | thin films | polymeric precursor | O$_3$ | 600 ppb | 250 | 170–580 | ~2/<5 min | 75 ppb | [289] |

PdO ultrathin films of thickness (5–10 nm) were grown on $SiO_2$/Si substrate by the thermal oxidation process. The PdO ultrathin film sensor provided a very high response value of approximately 1700 towards 100 ppm $O_3$ with superior low LOD (10 ppm), high signal stability and reproducibility at a moderate temperature of 175°C [279].

A nanostructured $In_2O_3$ thin film-based gas sensor prepared by sol-gel processes was found to be another suitable material for $O_3$ sensing. The sensor showed high sensitivity with a response of 20 for 400 ppm $O_3$, good reproducibility and selectivity against the other interfering gases at 100°C [272]. A photon stimulated ozone sensor based on $In_2O_3$ nanoparticles (7 nm) was fabricated by the MOCVD technique and the sensor was observed to be highly sensitive and stable in highly humid conditions for 10 ppm $O_3$ at room temperature [273].

The $In_2O_3$ microsphere (15 nm) based sensor has also been found very impressive with its fast response/recovery time of 60/40 s for 40 ppm $O_3$ at 150°C operating temperature. The sensor showed good selectivity, stability, reproducibility and linear response towards $O_3$ (40–240 ppm) with extremely low LOD of 10 ppm [275].

Nanostructured $O_3$ gas sensors decorated by noble metals Au, Cu, etc. [280,281] and transition metals Co, Fe, etc. [281] have become one of the effective approaches to improve gas sensing performance of $O_3$. In case of $O_3$, the selectivity i.e. detecting and quantifying pure $O_3$ from the mixture of various gases in the atmosphere at room temperature is a difficult task and NMOSs surfaces modified by noble metals have proved to be of great use to address this problem. An Au-decorated $TiO_2$ core–shell nanoparticle (9–12 nm) prepared by using the sol-gel method showed super-fast response/recovery times of 2/5 s towards (0.5–7.0) ppm $O_3$ at room temperature. This sensor exhibits excellent selectivity towards $O_3$ against the various gases such as $CH_3OH$, $C_2H_5OH$, $H_2$ and $NO_2$ [280]. The co-doped $SnO_2$ thin films sensors were grown on quartz substrates by the spray pyrolysis method and it was found that 4 wt% of cobalt nanoparticles enhanced the gas sensing response (approx. 10 for 1 ppm $O_3$) at 270°C [281].

The gas sensing performance of the NMOSs-based gas sensor can also be enhanced by amalgamating them with other metal oxides or carbon-based nanostructures [282]. For these MOS nanocomposites, the heterojunctions (the interfaces between different metal oxides) can accelerate the response speed of the sensor via effectively enhancing the electron transfer between different species. Besides the composite oxides, many mesoporous structures have been developed to improve the sensing parameters especially the sensitivity and response speed of these composite oxides-based sensors. The accumulation of nanoparticles to the composite oxides are beneficial for the adsorption and desorption of gas molecules, thus most sensors composed of the composite oxides exhibit very fast response times at room temperature [282].

The $In_2O$/$SiN_x$ thin film was grown on YX $LiTaO_3$ substrates using a sputtering technique and a SAW-based sensor was designed [282]. The sensor showed a large response (approx. 171) towards 40 ppm $O_3$ at 195°C. The sensor was found to perform very well as far as reversibility and repeatability were considered but long response/recovery time of 3/7.5 min made it unfit for practical application. The ZnO/$SnO_2$-based composite heterojunctions were prepared by using a hydrothermal method [277]. The composite heterojunction is composed of a ZnO needle with (100) orientation of the hexagonal phase of ZnO and $SnO_2$ nanoparticles with the rutile phase of $SnO_2$. The sensor displayed an extremely fast response/recovery time of 13/90 s for 0.3 ppm $O_3$ at room temperature, after UV exposure.

In addition, this sensor has very low LOD (0.3 ppm of $O_3$ with excellent sensing response, 37.5) and cross sensitivity/selectivity against the other interfering gases such as $NH_3$ (1 ppm), $NO_2$ (1 ppm) and CO (1 ppm).

The nanocomposites combining $WO_3$ nanoparticles (220 nm) and rGO nanosheets were fabricated on the surface of graphene nanosheets by a liquid flame spray method using $WCl_6$ and rGO as a precursor to enhance the gas sensing performance for $O_3$. This nanocomposite-based sensor exhibited very large response (approx. 370) towards $O_3$ as compared to pure rGO or $WO_3$ nanoparticles. The sensor showed the good response/recovery time of 17.1/32.7 s (for 0.3 wt% rGO) for 10 ppm $O_3$ at 150°C and also very low (0.5 ppm) LOD for the $O_3$ gas [284]. A single-walled carbon nanotubes/$SnO_2$-based $O_3$ gas sensor exhibited a fast response/recovery time with good sensitivity compared to a pure $SnO_2$-based sensor at room temperature. The CNT/$SnO_2$-based sensor has good LOD (1 ppm) of $O_3$ and remarkable stability (over one month) [283].

The gas sensing performance of $O_3$ gas sensors can also be improved by doping with metal ions in NMOSs owing to the increased number of active sites and defects on the surface of NMOS nanocrystals. These active sites enhance the amount of oxygen species and increase the adsorption of gas molecules on the sensor's surface. For the improvement in gas sensing performance of the NMOS, doping of metal ions such as $Zn^{2+}$, [285], $Co^{3+}$, [287,288] and $Fe^{3+}$, [251,289] has been reported in the literature.

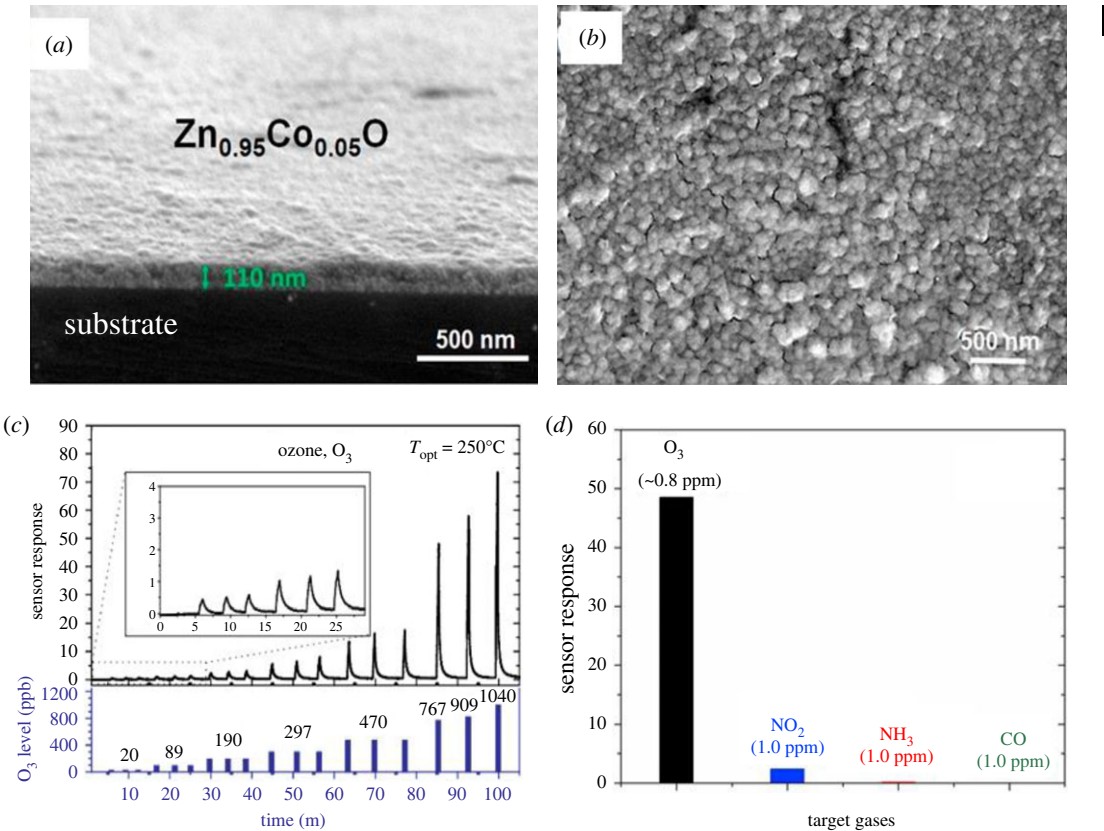

**Figure 10.** (*a*) SEM cross-sectional image, (*b*) surface views of $Zn_{0.95}Co_{0.05}O$ thin film, (*c*) ozone ($O_3$) sensor response of $Zn_{0.95}Co_{0.05}O$ thin film at 250°C exposed to different $O_3$ levels (20–1040 ppb) (inset shows a detailed region of sensor response towards 20 and 89 ppb of $O_3$ gas), and (*d*) comparison of the sensor responses of the $Zn_{0.95}Co_{0.05}O$ thin film exposed towards different gases at an operating temperature of 250°C. Reprinted with permission from [287].

The $Zn_{0.95}Co_{0.05}O$ thin film prepared by the spray pyrolysis technique revealed good sensitivity, repeatability and total reversibility towards $O_3$ gas of concentrations, 20–1040 ppb [287]. The SEM cross-sectional view and surface morphology of $Zn_{0.95}Co_{0.05}O$ thin film are shown in figure 10*a*,*b*, respectively. The $Co^{3+}$ improved the gas sensing of ZnO film owing to increase of the interaction sites for target gas. The ozone sensor response of $Zn_{0.95}Co_{0.05}O$ thin film at 250°C is shown in figure 10*c*. The sensor was found very selective for $O_3$ as tested for other gases; $NO_2$, $NH_3$, and CO as shown in figure 10*d*. The sensing performance was examined for 9 days and observed to be stable. Thus, the $Zn_{0.95}Co_{0.05}O$ thin film sensor can be used for practical applications as an $O_3$ gas sensor.

Excessive $O_3$ is a dangerous pollutant emitted from various industrial and household activities. In the past few years, the detection of $O_3$ gas has attracted a lot of interest of researchers. The above discussion enables us to conclude that versatile MOSs play a significantly important role in $O_3$ detection as well. Again, the metal oxide nanostructures such as core–shell structure, nanospheres and graphene oxide and rGO-based nanocomposites show faster response which is further enhanced by decorating them with noble metals. Thus, in $O_3$ detection the MOS materials are also promising candidates for commercial use.

Recently, some another methods are also being developed to detect the $O_3$ gas which includes: photoacoustic [293–296], optical $O_3$ gas sensors [297–299] $O_3$ gas sensors. Keeratirawee *et al*. [293] reported photoacoustic detection of $O_3$ gas with a red laser diode using the absorption band. The sensor exhibited about 1.6 ppm LOD for an optical power of the laser diode of about 130 mW. The optical $O_3$ sensor based on indium oxide nanoparticles were reported by Wang *et al*. [273]. This sensor showed low LOD of 10 ppb towards $O_3$ gas at room temperature. Wu *et al*. [298] prepared optical $O_3$ gas sensors by using $TiO_2$–$WO_3$ composites. The sensor exhibited a good response (23.8) and response/recovery times (155/235 s) towards 2.5 ppm $O_3$ at room temperature with LOD of 1000 ppb. This type of sensor has many disadvantages such as limited detection range (sensor dynamic range), cross sensitivity, weather effects (e.g. humidity), cost, light emitting diode (LED) as a light source and sensor robustness [300].

# 4. Conclusion

The climate is changing and affecting the various regions around the world. In future, its negative impacts will most likely be much more severe than before. The huge challenge before us is to first stabilize and then reduce the anthropogenic GHG emissions. To overcome this problem, the very first step in this direction is to measure/detect the GHGs emissions precisely in the Earth's atmosphere and for this purpose, we must have reliable measurement systems for the GHGs. The related data acquisition techniques demand innovative, cost effective, robust and accurate sensors. However, the area of sensors is emerging swiftly of which NMOS-based sensors are paid the utmost attention because of the fulfilment of all these requirements. This review article presents a comprehensive study of the role of NMOS-based gas sensors in the detection of GHGs and their potential to be used in commercial applications. The NMOS-based gas sensors are found to be highly effective in the detection of various GHGs in the Earth's atmosphere. A wide range of morphologies such as nanorods, nanofibres, nanosheets, nanospheres and core–shell structures, etc. are exhibited by nanostructured metal oxides with an advantage of higher surface area and results in increment in adsorption sites available for sensing GHGs. The nanostructured composites with novel materials such as graphene, graphene oxide and rGO make them highly promising candidates for gas sensing applications. Further improvement in their sensing characteristics can be made by doping of noble metals such as Au, Pt and Pd in NMOS. A variety of simple and economical methods are available for the synthesis of NMOS-based nanomaterials which facilitates the tuning of sensing parameters as well as commercial use. Many challenges such as reduction in operating temperature, high selectivity, stability in different measuring environments and longer life time to design suitable NMOS-based GHGs sensors are still open for the enthusiastic researchers in this field.

Data accessibility. This article has no additional data.

Authors' contributions. Y.K.G.: contributed towards the conception and design of the review, acquisition of data, analysis and interpretation of data, drafting of the article. K.S.: contributed towards the conception and design of this study, and drafting of the article. S.T.: helped in the acquisition of data, drafting of the manuscript and arranging the references using Mendley software. A.K.A.: supplied the acquisition of data, analysis and interpretation of data. M.C.: worked on the acquisition of data and the drafting of the manuscript. B.P.S.: provided the conception and design of this review article, added important intellectual content and coordinated to draft the final version of the manuscript. It is hereby declared that all the authors gave approval for the publication of this review article in Royal Society Open Science.

Competing interests. We declare we have no competing interests

Funding. We received no funding for this study.

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
