## [Peer Review File · Royal Society Open Science]

Review History

RSOS-201324.R0 (Original submission)

Review form: Reviewer 1

Is the manuscript scientifically sound in its present form?

No

Are the interpretations and conclusions justified by the results?

No

Is the language acceptable?

No

Do you have any ethical concerns with this paper?

No

Have you any concerns about statistical analyses in this paper?

No

Recommendation?

Major revision is needed (please make suggestions in comments)

Comments to the Author(s)

Comments

The abstract

1)It presents nitrogen oxides (NO_x) as a greenhouse gas together with carbon dioxide (CO₂) and methane (CH₄). You can't use it, because only nitrous oxide (N₂O) is a greenhouse gas or other oxides of nitrogen (N₀, NO₂ and NO₃) are not greenhouses. This generalization cannot be done. See references for reliable sources (greenhouse gases) see below:

a)EPA - US (Environmental Protection Agency)

<https://www.epa.gov/ghgemissions/overview-greenhouse-gases>

Gases estufa: (CO₂), (CH₄), (N₂O) and Fluorinated Gases

b) (IPCC) The Intergovernmental Panel on Climate Change - ONU.

<https://www.ipcc.ch/>

2) When only the title of the article is read, we have the impression of a review on greenhouse gas sensors based on metallic oxide semiconductor (MOS).

But the article has lost its focus, and makes further revisions on the achievements of climate change and its solutions (renewable energies). These themes could be approached as a justification and motivation for the article. But it has an excess of pages, sometimes in a simple language but more suitable for a book than for a scientific text (article).

The review (MOS) does not mention other gas detection techniques such as photoacoustic spectroscopy and chemiluminescence etc ... There could be a comparison of the advantages and disadvantages with other techniques. The NO₂ gas sensors (which is not a greenhouse) are widely discussed, so it is beyond the scope of the article. The greenhouse gas that is (N₂O), has no comment in the text, nor reference !! .And the article still lacked comments on the part of the sensors (MOS).

3)"The article needs a new writing that defines its objectives well"

4)Nothing is said about other GHC sensors, only satellite sensing. Why ? there are many gas detection techniques . (The introduction)

5)Let's see the statements in the abstract

a) "GHGs sensors and their sensing mechanism have been discussed briefly"

This should be better discussed, rather than discussing details of climate change.

b)'The status of global GHGs emissions during the COVID-19 pandemic lockdown has also been discussed'.

Why discuss this topic in the article? I don't see this essential topic for a sensor review.

The introduction

1)This topic is about environmental sustainability, this topic is really important, but it will be for this article.

In the paragraphs below !!

- quality of life in terms of threatening water supplies, compromising ecosystems and impeding growth for future generations. The environment around us is an essential part of human survival. The more we don't care about our environment, the more it will become intoxicated with contaminants and toxins that have a harmful impact on our lives in numerous ways

- "In particular, the pace of global economic growth during the past decades, paved the way to a continuous decline in the availability of natural resources such as forests (cut down for agriculture/demand for wood), air/land/water pollution. This in turn imposes severe health problems, damage to the productivity of land and seas, sources of oil/coal/gas, loss of fishing stocks due to overfishing and loss of species diversity etc. to the mankind. Our environment is composed of some basic components such as water, air, soil, flora, fauna, and human beings and hence economic development must move forward with the stability between these components"

2) See the statements below

a) On satellite observations as an additional or alternative tool for the detection and monitoring of anthropogenic GHGs emissions such as CO₂, CH₄, NO_x etc

b) NMOS-based sensors have been shown to be sensitive to a large range of GHGs, mainly to CO₂, CH₄, NO_x and O₃,

c) It is noteworthy here that the present review centred around only on the performance analysis of available NMOS GHGs (CO₂, CH₄, NO_x, O₃ etc.) sensors

NO_x !!!! it's greenhouse gas?

In the section 'Concentration of GHGs in earth's environment'

In the first paragraph - Look, nitrous oxide (N₂O), greenhouse gas

'The primary GHGs in the atmosphere are water vapor, carbon dioxide, methane, nitrous oxide, and ozone'.

3) Concentração de GEEs no meio ambiente da Terra

Table 1 shows NO_x as a greenhouse, in fact it is N₂O

The reference in Table 1 is (28) being the same reference as in Figure 2,

Both table and graph show 2000 data

Shows the figure 2. data 20 years ago. Reference below

https://commons.wikimedia.org/wiki/File:Greenhouse_gas_by_sector_2000.svg

Reference 28 is presented in the article, where the consultation was made 2020, but data are very old.

4) It has a section 3.3 MOS-based NO_x gas sensors, entirely dedicated to NO₂ gas, which is not a greenhouse! It is a dangerous gas for human health, it generates acid rain and attacks the ozone layer. But the title of the article is sensors for greenhouse gases. Nothing is talked about sensors for N₂O gas

5) It has a section 4. Status of global GHGs emissions during the COVID -19 pandemic lockdown. The theme is important (of course), but it may not be for the central theme of the article (gas sensors)

6)The conclusions must be completely rewritten, according to the objectives of the work (focus). It is certainly not renewable energy or the consequences of climate change.

Review form: Reviewer 2 (Nitin Shelke)

Is the manuscript scientifically sound in its present form?

Yes

Are the interpretations and conclusions justified by the results?

Yes

Is the language acceptable?

Yes

Do you have any ethical concerns with this paper?

No

Have you any concerns about statistical analyses in this paper?

No

Recommendation?

Accept with minor revision (please list in comments)

Comments to the Author(s)

The manuscript can be accept with minor revision.

Decision letter (RSOS-201324.R0)

Dear Dr Sharma:

Title: Nanostructured Metal Oxide Semiconductor-based Sensors for Greenhouse Gas Detection: Progress and Challenges
Manuscript ID: RSOS-201324

The editor assigned to your manuscript has now received comments from reviewers. We would like you to revise your paper in accordance with the referee and Subject Editor suggestions which can be found below (not including confidential reports to the Editor). Please note this decision does not guarantee eventual acceptance.

Please submit your revised paper before 31-Jan-2021. Please note that the revision deadline will expire at 00.00am on this date. If we do not hear from you within this time then it will be assumed that the paper has been withdrawn. In exceptional circumstances, extensions may be possible if agreed with the Editorial Office in advance. We do not allow multiple rounds of revision so we urge you to make every effort to fully address all of the comments at this stage. If deemed necessary by the Editors, your manuscript will be sent back to one or more of the original reviewers for assessment. If the original reviewers are not available we may invite new reviewers.

On behalf of the Subject Editor Professor Anthony Stace and the Associate Editor Dr Dattatray Late.

RSC Associate Editor:
Comments to the Author:
Major Revision needed.

RSC Subject Editor:
Comments to the Author:
(There are no comments.)

Reviewers' Comments to Author:
Reviewer: 1
Comments to the Author(s)
Comments

The abstract

1) It presents nitrogen oxides (NO_x) as a greenhouse gas together with carbon dioxide (CO₂) and methane (CH₄). You can't use it, because only nitrous oxide (N₂O) is a greenhouse gas or other oxides of nitrogen (N₀, NO₂ and NO₃) are not greenhouses. This generalization cannot be done. See references for reliable sources (greenhouse gases) see below:

a) EPA - US (Environmental Protection Agency)

<https://www.epa.gov/ghgemissions/overview-greenhouse-gases>

Gases estufa: (CO₂), (CH₄), (N₂O) and Fluorinated Gases

b) (IPCC) The Intergovernmental Panel on Climate Change - ONU.

<https://www.ipcc.ch/>

2) When only the title of the article is read, we have the impression of a review on greenhouse gas sensors based on metallic oxide semiconductor (MOS).

But the article has lost its focus, and makes further revisions on the achievements of climate change and its solutions (renewable energies). These themes could be approached as a justification and motivation for the article. But it has an excess of pages, sometimes in a simple language but more suitable for a book than for a scientific text (article).

The review (MOS) does not mention other gas detection techniques such as photoacoustic spectroscopy and chemiluminescence etc ... There could be a comparison of the advantages and disadvantages with other techniques. The NO₂ gas sensors (which is not a greenhouse) are widely discussed, so it is beyond the scope of the article. The greenhouse gas that is (N₂O), has no comment in the text, nor reference !! .And the article still lacked comments on the part of the sensors (MOS).

3) "The article needs a new writing that defines its objectives well"

4) Nothing is said about other GHC sensors, only satellite sensing. Why ? there are many gas detection techniques . (The introduction)

5) Let's see the statements in the abstract

a) "GHGs sensors and their sensing mechanism have been discussed briefly"

This should be better discussed, rather than discussing details of climate change.

b) 'The status of global GHGs emissions during the COVID-19 pandemic lockdown has also been discussed'.

Why discuss this topic in the article? I don't see this essential topic for a sensor review.

The introduction

1) This topic is about environmental sustainability, this topic is really important, but it will be for this article.

In the paragraphs below !!

- quality of life in terms of threatening water supplies, compromising ecosystems and impeding growth for future generations. The environment around us is an essential part of human survival. The more we don't care about our environment, the more it will become intoxicated with contaminants and toxins that have a harmful impact on our lives in numerous ways

- "In particular, the pace of global economic growth during the past decades, paved the way to a continuous decline in the availability of natural resources such as forests (cut down for agriculture/demand for wood), air/land/water pollution. This in turn imposes severe health problems, damage to the productivity of land and seas, sources of oil/coal/gas, loss of fishing stocks due to overfishing and loss of species diversity etc. to the mankind. Our environment is composed of some basic components such as water, air, soil, flora, fauna, and human beings and hence economic development must move forward with the stability between these componen"t

2)See the statements below

a)On satellite observations as na additional or alternative tool for the detection and monitoring of anthropogenic GHGs emissions such as CO₂, CH₄, NO_x etc

b)NMOS-based sensors have been shown to be sensitive to a large range of GHGs, mainly to CO₂, CH₄, NO_x and O₃,

c)It is noteworthy here that the presente review centred around only on the performance analysis of available NMOS GHGs (CO₂, CH₄, NO_x, O₃ etc.) sensors

NO_x !!!! it's greenhouse gas?

In the section 'Concentration of GHGs in earth's environment"

In the first paragraph - Look, nitrous oxide (N₂O), greenhouse gas

'The primary GHGs in the atmosphere are water vapor, carbon dioxide, methane, nitrous oxide, and ozone'.

3)Concentração de GEEs no meio ambiente da Terra

Table 1 shows NO_x as a greenhouse, in fact it is N₂O

The reference in Table 1 is (28) being the same reference as in Figure 2,

Both table and graph show 2000 data

Shows the figure 2. data 20 years ago. Reference below

https://commons.wikimedia.org/wiki/File:Greenhouse_gas_by_sector_2000.svg

Reference 28 is presented in the article, where the consultation was made 2020, but data are very old.

4)It has a section 3.3 MOS-based NO_xgas sensors, entirely dedicated to NO₂ gas, which is not a greenhouse! It is a dangerous gas for human health, it generates acid rain and attacks the ozone layer. But the title of the article is sensors for greenhouse gases. Nothing is talked about sensors for N₂O gas

5) It has a section 4. Status of global GHGs emissions during the COVID -19 pandemic lockdown. The theme is important (of course), but it may not be for the central theme of the article (gas sensors)

6)The conclusions must be completely rewritten, according to the objectives of the work (focus). It is certainly not renewable energy or the consequences of climate change.

Reviewer: 2

Comments to the Author(s)

The manuscript can be accepted with minor revision.

Author's Response to Decision Letter for (RSOS-201324.R0)

See Appendix A.

Decision letter (RSOS-201324.R1)

Dear Dr Sharma:

Title: Nanostructured Metal Oxide Semiconductor-based Sensors for Greenhouse Gas Detection: Progress and Challenges
Manuscript ID: RSOS-201324.R1

It is a pleasure to accept your manuscript in its current form for publication in Royal Society Open Science. The chemistry content of Royal Society Open Science is published in collaboration with the Royal Society of Chemistry.

On behalf of the Subject Editor Professor Anthony Stace and the Associate Editor Dr Dattatray Late.

RSC Associate Editor
Comments to the Author:
Accept as is

Reviewer(s)' Comments to Author:

Appendix A

“Response letter to Referees” for the manuscript ID: RSOS-201324 “Nanostructured Metal Oxide Semiconductor-based Sensors for Greenhouse Gas Detection: Progress and Challenges”

Dear Editor,

We are greatly thankful to you and reviewers for the valuable comments and suggestions on our research article to improve the quality of manuscript entitled “**Nanostructured Metal Oxide Semiconductor-based Sensors for Greenhouse Gas Detection: Progress and Challenges**” (RSOS-201324). We have revised the manuscript in the light of reviewer’s comments/suggestions. Kindly, find the attached revised manuscript for your consideration. All the modified/revised part has marked in yellow in the revised manuscript. The point-wise clarifications/respond to the comments made by the referees are given below. We hope this will also be acceptable to the reviewer and editor.

With kind regards,

Dr. Kavita Sharma
Chaudhary Charan Singh University
Meerut, India

Response to reviewer comment

Reviewer: 1(Major Revision)

The abstract

Comment 1: It presents nitrogen oxides (NO_x) as a greenhouse gas together with carbon dioxide (CO₂) and methane (CH₄). You can't use it, because only nitrous oxide (N₂O) is a greenhouse gas or other oxides of nitrogen (NO, NO₂ and NO₃) are not greenhouses. This generalization cannot be done.

See references for reliable sources (greenhouse gases) see below:

a)EPA – US (Environmental Protection Agency)

<https://www.epa.gov/ghgemissions/overview-greenhouse-gases>

Gases estufa: (CO₂), (CH₄), (N₂O) and Fluorinated Gases

b) (IPCC) The Intergovernmental Panel on Climate Change – ONU.

<https://www.ipcc.ch/>

Response: We thank the reviewer for raising this important comment on (NO_x). We completely agree with the reviewer. In the revised manuscript, we have replaced NO_x with N₂O as per the suggestion of learned reviewer through the links suggested in a) and b) and also cite the link in reference list. Now the greenhouse gases referred in revised manuscript are; CO₂, CH₄, N₂O and O₃.

Comment 2: When only the title of the article is read, we have the impression of a review on greenhouse gas sensors based on metallic oxide semiconductor (MOS).

But the article has lost its focus, and makes further revisions on the achievements of climate change and its solutions (renewable energies). These themes could be approached as a

justification and motivation for the article. But it has an excess of pages, sometimes in a simple language but more suitable for a book than for a scientific text (article).

The review (MOS) does not mention other gas detection techniques such as photoacoustic spectroscopy and chemiluminescence etc. There could be a comparison of the advantages and disadvantages with other techniques. The NO₂ gas sensor (which is not a greenhouse gas) is widely discussed, so it is beyond the scope of the article. The greenhouse gas that is (N₂O), has no comment in the text, nor reference !!. And the article still lacked comments on the part of the sensors (MOS).

Response: We agree with the reviewer. We have removed the excess of pages on the achievements of climate change and its solutions (renewable energies) in the revised manuscript accordingly.

As per reviewer's suggestion, authors have mentioned other gas detection techniques such as photoacoustic spectroscopy, chemiluminescence, electrochemical sensor, thermometric sensors, catalytic sensor, gas chromatography, chemical sensor and mass resistive sensor in the revised manuscript.

Authors have given a comparison of the advantages and disadvantages of MOS-based technique with other techniques as per the suggestion of reviewer and highlighted by yellow colour in Table-2 of the revised manuscript.

Authors are very much thankful to the reviewer for the comment on the NO₂ gas sensors. We agree with the reviewer. As NO₂ is not a greenhouse gas. Therefore, authors have removed the given content for NO₂ gas sensors.

In the revised manuscript, authors have included a wide discussion about the various studies/reports available on MOS-based sensors for N₂O (greenhouse gas). Authors have also introduced the other recent available techniques for N₂O detection as per the reviewer suggestion/expectation.

Comment 3: "The article needs a new writing that defines its objectives well"

Response: We are agreeing with the reviewer. In the revised manuscript, authors have rewritten the objectives of this review and highlighted that with yellow colour.

Comment 4: Nothing is said about other GHG sensors, only satellite sensing. Why? there are many gas detection techniques. (The introduction)

Response: Although satellite sensing is an emerging interest of climate scientists in order to have a wide spatial and temporal coverage of GHGs emission globally. But authors agree completely with the reviewer's concern on the theme of this review and therefore authors have removed satellite sensing paragraph in the introduction of revised manuscript. Instead other GHG gas sensors have been included in introduction of revised manuscript and highlighted in yellow colour.

Comment 5: Let's see the statements in the abstract;

a) "GHGs sensors and their sensing mechanism have been discussed briefly"
This should be better discussed, rather than discussing details of climate change.

Response: We agree with the reviewer. In the revised manuscript, authors have rewritten the abstract as per the suggestion of the reviewer and highlight the rewrite part with yellow color.

b) "The status of global GHGs emissions during the COVID-19 pandemic lockdown has also

been discussed’.

Why discuss this topic in the article? I don't see this essential topic for a sensor review.

Response: We agree with the reviewer. Authors have discussed the status of global GHGs emissions during the COVID-19 pandemic lockdown because of the interest and motivation followed by the reduction of global GHGs emissions level.

However, authors have removed this section 4 in the revised manuscript as per the reviewer suggestion.

Reviewer: 2 (Minor revision)

Comments to the Author(s)

The introduction

Comment 1: This topic is about environmental sustainability, this topic is really important, but it will be for this article.

In the paragraphs below !!

- quality of life in terms of threatening water supplies, compromising ecosystems and impeding growth for future generations. The environment around us is an essential part of human survival. The more we don't care about our environment, the more it will become intoxicated with contaminants and toxins that have a harmful impact on our lives in numerous ways

- “In particular, the pace of global economic growth during the past decades, paved the way to a continuous decline in the availability of natural resources such as forests (cut down for agriculture/demand for wood), air/land/water pollution. This in turn imposes severe health problems, damage to the productivity of land and seas, sources of oil/coal/gas, loss of fishing stocks due to overfishing and loss of species diversity etc. to the mankind. Our environment is composed of some basic components such as water, air, soil, flora, fauna, and human beings and hence economic development must move forward with the stability between these componen”t.

Response: We agree with the reviewer. In the revised manuscript, authors have removed all the above said statements/paragraphs about the environment sustainability and modified part of introduction is highlighted by yellow colour as per the reviewer's suggestion.

Comment 2: See the statements below;

a) On satellite observations as na additional or alternative tool for the detection and monitoring of anthropogenic GHGs emissions such as CO₂, CH₄, NO_x etc

b) NMOS-based sensors have been shown to be sensitive to a large range of GHGs, mainly to CO₂, CH₄, NO_x and O₃,

c) It is noteworthy here that the present review centred around only on the performance analysis of available NMOS GHGs (CO₂, CH₄, NO_x, O₃ etc.) sensors

NO_x !!!! it's greenhouse gas?

In the section ‘Concentration of GHGs in earth’s environment’

In the first paragraph - Look, nitrous oxide (N₂O), greenhouse gas

‘The primary GHGs in the atmosphere are water vapor, carbon dioxide, methane, nitrous oxide, and ozone’.

Response: We agree with the reviewer as N₂O is a greenhouse gas. In the revised manuscript, authors have modified all the above said statements/paragraphs as per the reviewer suggestions.

Comment 3: Concentração de GEEs no meioambiente da Terra

Table 1 shows NO_x as a greenhouse, in fact it is N₂O

The reference in Table 1 is (28) being the same reference as in Figure 2,

Both table and graph show 2000 data

Shows the figure 2. data 20 years ago. Reference below

https://commons.wikimedia.org/wiki/File:Greenhouse_gas_by_sector_2000.svg

Reference 28 is presented in the article, where the consultation was made 2020, but data are very old.

Response: We agree with the reviewer. We have mentioned N₂O in place of NO_x in Table-1, in the revised manuscript as per reviewer suggestions.

Authors are very much thankful to the reviewer for the valuable comment on the reference (28) (reference no. 20 in the revised manuscript) in Table 1 and in Figure 2.

In the revised manuscript, we have modified the Table 1 and Figure 2 as per the data available in the references [(a)EPA – US (Environmental Protection Agency) <https://www.epa.gov/ghgemissions/overview-greenhouse-gases> Gases estufa: (CO₂), (CH₄), (N₂O) and Fluorinated Gases and (b) (IPCC) The Intergovernmental Panel on Climate Change – ONU. <https://www.ipcc.ch/>] suggested by reviewer 1. Authors also included the suggested reference in the reference list. Please note that all changes have been highlighted by yellow colour in revised manuscript.

Comment 4: It has a section 3.3 MOS-based NO_x gas sensors, entirely dedicated to NO₂ gas, which is not a greenhouse! It is a dangerous gas for human health, it generates acid rain and attacks the ozone layer. But the title of the article is sensors for greenhouse gases. Nothing is talked about sensors for N₂O gas.

Response: We agree with the reviewer. In the revised manuscript, authors have replaced complete section 3.3: MOS-based NO_x gas sensors with MOS-based N₂O gas sensors as suggested by the reviewer.

Comment 5: It has a section 4. Status of global GHGs emissions during the COVID -19 pandemic lockdown. The theme is important (of course), but it may not be for the central theme of the article (gas sensors).

Response: We agree with the reviewer. Authors have removed the “section 4: Status of global GHGs emissions during the COVID -19 pandemic lockdown” in the revised manuscript accordingly.

Comment 6: The conclusions must be, according to the objectives of the work (focus). It is certainly not renewable energy or the consequences of climate change.

Response: We are agreeing with the reviewer. In the revised manuscript, authors have completely rewritten the conclusions according to the objectives of the work as per reviewer suggestion.

In the last, authors are very much thankful to the reviewers for critically reviewing our review article and accepting the manuscript for publication with the major/minor revision.